# Nup98-dependent transcriptional memory is established independently of transcription

**Pau Pascual-Garcia†, Shawn C Little, Maya Capelson\***

Department of Cell and Developmental Biology, Penn Epigenetics Institute, Perelman School of Medicine, University of Pennsylvania, Philadelphia, United States

**Abstract** Cellular ability to mount an enhanced transcriptional response upon repeated exposure to external cues is termed transcriptional memory, which can be maintained epigenetically through cell divisions and can depend on a nuclear pore component Nup98. The majority of mechanistic knowledge on transcriptional memory has been derived from bulk molecular assays. To gain additional perspective on the mechanism and contribution of Nup98 to memory, we used single-molecule RNA FISH (smFISH) to examine the dynamics of transcription in *Drosophila* cells upon repeated exposure to the steroid hormone ecdysone. We combined smFISH with mathematical modeling and found that upon hormone exposure, cells rapidly activate a low-level transcriptional response, but simultaneously begin a slow transition into a specialized memory state characterized by a high rate of expression. Strikingly, our modeling predicted that this transition between non-memory and memory states is independent of the transcription stemming from initial activation. We confirmed this prediction experimentally by showing that inhibiting transcription during initial ecdysone exposure did not interfere with memory establishment. Together, our findings reveal that Nup98's role in transcriptional memory is to stabilize the forward rate of conversion from low to high expressing state, and that induced genes engage in two separate behaviors – transcription itself and the establishment of epigenetically propagated transcriptional memory.

**\*For correspondence:**
capelson@pennmedicine.upenn.
edu

**Present address:** †Center for Genomic Regulation (CRG), The Barcelona Institute of Science and Technology, Barcelona, Spain

**Competing interest:** The authors declare that no competing interests exist.

## Editor's evaluation

The authors quantitatively assess transcriptional memory in the context of mathematical modeling and testing of the models through single cell approaches. They extend their work to show how single cell data relates to population-level transcription outcomes. The models produced make predictions that the authors successfully test to demonstrate that transcription initiation is not necessary for establishment of memory.

## Introduction

Organisms continuously adapt to environmental changes, with transcriptional responses to varying stimuli playing critical roles in development and survival. One mechanism of cellular adaptation relies on the cells' ability to mount increasingly robust transcriptional responses upon repeated exposure to signals they have previously encountered. This phenomenon by which cells increase their re-activation kinetics after a period of repression is called transcriptional memory (*Avramova, 2015*; *Bonifer and Cockerill, 2017*; *D'Urso and Brickner, 2017*; *Fabrizio et al., 2019*). The ability of cells to increase transcriptional responsiveness upon subsequent exposure (i.e. 'remember' their previous exposure) can be propagated through cell division, and is thus considered epigenetic. Transcriptional memory is well-characterized for genes that respond to environmental changes: for example, in budding yeast,

rounds of inositol starvation trigger the transcriptional memory response for *INO1*, which encodes the enzyme catalyzing the limiting step in the biosynthesis of inositol (*Ahmed et al., 2010*; *Brickner et al., 2007*; *Light et al., 2010*); or in plants, where pre-exposure to various abiotic stresses such as drought or changes in salinity intensifies their resistance by elevating transcript levels from a subset of stress-response genes (*Ding et al., 2013*; *Ding et al., 2012*; *Lämke et al., 2016*; *Liu et al., 2016*; *Sani et al., 2013*). In both cases, transcriptional memory provides an advantage to survive environmental challenges by mounting a more robust transcriptional response that facilitates adjustment to new conditions.

Importantly, the primed state that is induced after initial exposure is not unique to environmental signals and can be found in different cellular processes. In mammalian cells, interferon-induced transcriptional memory is thought to confer an enhanced immune response to infectious agents (*Gialitakis et al., 2010*; *Kamada et al., 2018*; *Light et al., 2013*), and in *Drosophila*, transcriptional memory has been reported in the context of signaling by the steroid hormone ecdysone. Ecdysone or 20-hydroxyecdysone (20E) and its nuclear hormone receptor complex EcR/USP coordinate a transcriptional response that is essential for multiple events during fly development (*Hill et al., 2013*). A defined set of early ecdysone-induced genes, such as *ecdysone-induced protein 74EF* (*E74*) and *early gene at 23* (*E23*), are activated by the hormone receptor and remain primed for enhanced expression upon a second exposure for at least 24 hr (*Pascual-Garcia et al., 2017*). Further evidence of roles for transcriptional memory during organismal development comes from a study of re-activation dynamics during *Drosophila* embryogenesis, where transgenes were shown to re-activate more frequently and more rapidly after cell division if they were transcribing in the previous cell cycle (*Ferraro et al., 2016*).

Mechanistically, several models have been put forward to explain how transcriptional memory may work. One such model proposed that memory is based primarily on the cytoplasmic inheritance of a key regulatory factor, which is build up during the initial exposure and remains at high levels through the period of memory (*Kundu and Peterson, 2009*). Such a mode of inheritance regulates the long-term transcriptional memory of the yeast *GAL1* gene, which is repressed in normal glucose-containing media, but can be activated by switching cells to galactose (*Brickner et al., 2007*; *Kundu et al., 2007*; *Zacharioudakis et al., 2007*). In this case, the high expression of GAL1 regulators during initial activation and their subsequent persistence are thought to underlie the higher re-activation dynamics upon second exposure to galactose (*Kundu and Peterson, 2010*; *Sood et al., 2017*; *Zacharioudakis et al., 2007*). Another set of models focuses on a nuclear mode of inheritance of a transcriptional state, with chromatin and architectural features of the gene as the functional memory marks (*D'Urso and Brickner, 2017*; *Kundu and Peterson, 2009*; *Randise-Hinchliff and Brickner, 2018*). This model of transcriptional memory centers on the binding of gene-specific transcriptional factors to cis-acting DNA elements and on contacts of the gene with nuclear pore complex (NPC) components (or Nups). Significant efforts have been made to understand how transcriptional memory operates for the yeast *INO1* gene, where recruitment of transcriptional factor Sfl1 at *INO1* promoter and contact with Nup100 (homolog of metazoan Nup98) have been shown to promote the incorporation of histone variant H2A.Z and H3K4me2, both of which have been linked to memory establishment in multiple systems (*Bevington et al., 2016*; *Brickner et al., 2007*; *D'Urso et al., 2016*; *Gialitakis et al., 2010*; *Lämke et al., 2016*; *Light et al., 2013*; *Light et al., 2010*; *Muramoto et al., 2010*; *Petter et al., 2011*).

Studies in multiple organisms have revealed that Nup98 is an evolutionarily conserved factor necessary for transcriptional memory. In addition to INO1 regulation described above, interferon-induced genes require Nup98 for memory in human cells (*Light et al., 2013*), where Nup98 similarly promotes deposition of the H3K4Me2 mark. However, this memory event occurs at genes in the nuclear interior, which reflects the reported ability of Nup98 and other Nups to move on and off the pore and occupy intranuclear locations in metazoan cells (*Capelson et al., 2010*; *Kalverda et al., 2010*). Nup98 is also necessary for transcriptional memory in ecdysone-induced signaling. In *Drosophila* embryonic cells, loss of Nup98 does not affect transcription of ecdysone-induced genes during initial ecdysone exposure, but results in a poor memory response during re-induction (*Pascual-Garcia et al., 2017*). In fly cells, Nup98 has been implicated in the maintenance of ecdysone-induced enhancer-promoter loops of genes like *E74* and *E23*, providing evidence that Nups can influence transcriptional memory through changes in chromatin contacts (*Pascual-Garcia et al., 2017*). Thus, the roles of NPC components in transcriptional memory appear complex and may involve a network of coordinated events.

Investigation of this conserved function of Nup98 provides an opportunity to build further understanding of the memory phenomenon.

Previous work on transcriptional memory has primarily relied upon population-level analyses, using bulk biochemical and molecular assays to investigate its mechanisms (*Brickner et al., 2007*; *Ding et al., 2012*; *Gialitakis et al., 2010*; *Pascual-Garcia et al., 2017*). Here, we aimed to understand how transcriptional memory could work at the level of single cells and how single-cell behaviors can give rise to population-level transcriptional outcomes. We combined precise quantification of ecdysone-induced *E74* mRNAs by single-molecule RNA FISH (smFISH) with mathematical modeling to describe transcriptional memory and the role of Nup98 from a single-cell perspective. By comparing predicted population distributions based on our modeling approaches to actual distributions of nascent transcriptional states obtained by smFISH, we have defined the transcriptional parameters that change in the reinduced vs. induced states and shed light on the contribution of Nup98 to the transcriptional process. Strikingly, we found that transition into the memory state is independent of the extent of transcriptional activity during initial induction. Our results introduce a possible model that accounts for cell- and population-level dynamics during transcriptional memory response, allow us to rule out some previously proposed models for ecdysone-mediated memory, and suggest a functional role of Nup98 in driving the formation of a memory state that is separate from ongoing transcription.

## Results

### Assessing Nup98-dependent transcriptional memory in absolute units

We have previously shown that ecdysone-inducible genes exhibit transcriptional memory, and that Nup98 is required for the proper establishment and/or maintenance of the primed memory state (*Pascual-Garcia et al., 2017*). To further investigate the role of Nup98 in modulating transcription, we examined the transcription dynamics of the ecdysone-responsive gene *E74* during repeated hormone exposure in more detail. *Drosophila* S2 cells were exposed to synthetic 20E for a 4 hr initial induction, then washed and allowed to recover for 24 hr before re-exposure to 20E for an additional 4 hr (*Figure 1A*). We assayed *E74* expression dynamics during both exposures by collecting cells every 30 min, isolating mRNA, and performing RT-qPCR on cells treated with dsRNA against the control *white* gene (dsWhite) or against Nup98 (dsNup98). To aid in quantitative analysis, we adapted a previously described protocol (*Petkova et al., 2014*) to estimate the absolute number of *E74* mRNA molecules per cell. We performed RT-qPCR using known numbers of in vitro transcribed RNAs as template spanning six orders of magnitude. This allowed us to construct a calibration curve relating the qPCR threshold cycle Ct to the number of input molecules (*Figure 1—figure supplement 1A*; see Materials and methods). We measured the number of cells from which we extracted mRNA for each experimental condition and time point. We also estimated the mRNA recovery efficiency of the mRNA extraction step. Together, these measurements allowed us to convert experimental Ct values into absolute numbers of *E74* mRNA molecules per cell, generating a detailed description of *E74* expression dynamics during hormone exposure in control and Nup98 knockdown cells (*Figure 1B*).

Using this approach, we found as expected that the number of *E74* mRNA molecules per cell reaches considerably higher amounts during re-induction than during initial induction in control dsWhite-treated cells (*Figure 1B*). *E74* transcripts accumulate slowly during the first 2 hr of initial hormone exposure before increasing. Moreover, the accumulation trajectory in Nup98-depleted cells is not different from control during the first induction. In contrast, accumulation is rapidly onset during the second induction in control cells, whereas in Nup98 knockdown conditions, the second induction is more similar to the first induction (*Figure 1B*). This supports the proposed requirement for Nup98 in transcriptional memory, in agreement with previous work from our and other laboratories (*Light et al., 2013*; *Pascual-Garcia et al., 2017*). The efficiency of Nup98 depletion was similar during both the first and second inductions (*Figure 1—figure supplement 1B*), supporting our conclusion that the response of *E74* to initial exposure does not require normal levels of Nup98; instead, cells require Nup98 to rapidly express *E74* upon repeated exposure.

In the simplest model of hormone-induced transcription, all loci would be rapidly activated upon hormone treatment and then begin producing mRNAs at a constant rate. If true, then mRNA content should be relatively uniform between cells, particularly at later times after induction, when cells will have ample opportunity for their expression levels to closely approach the mean level. To ask whether

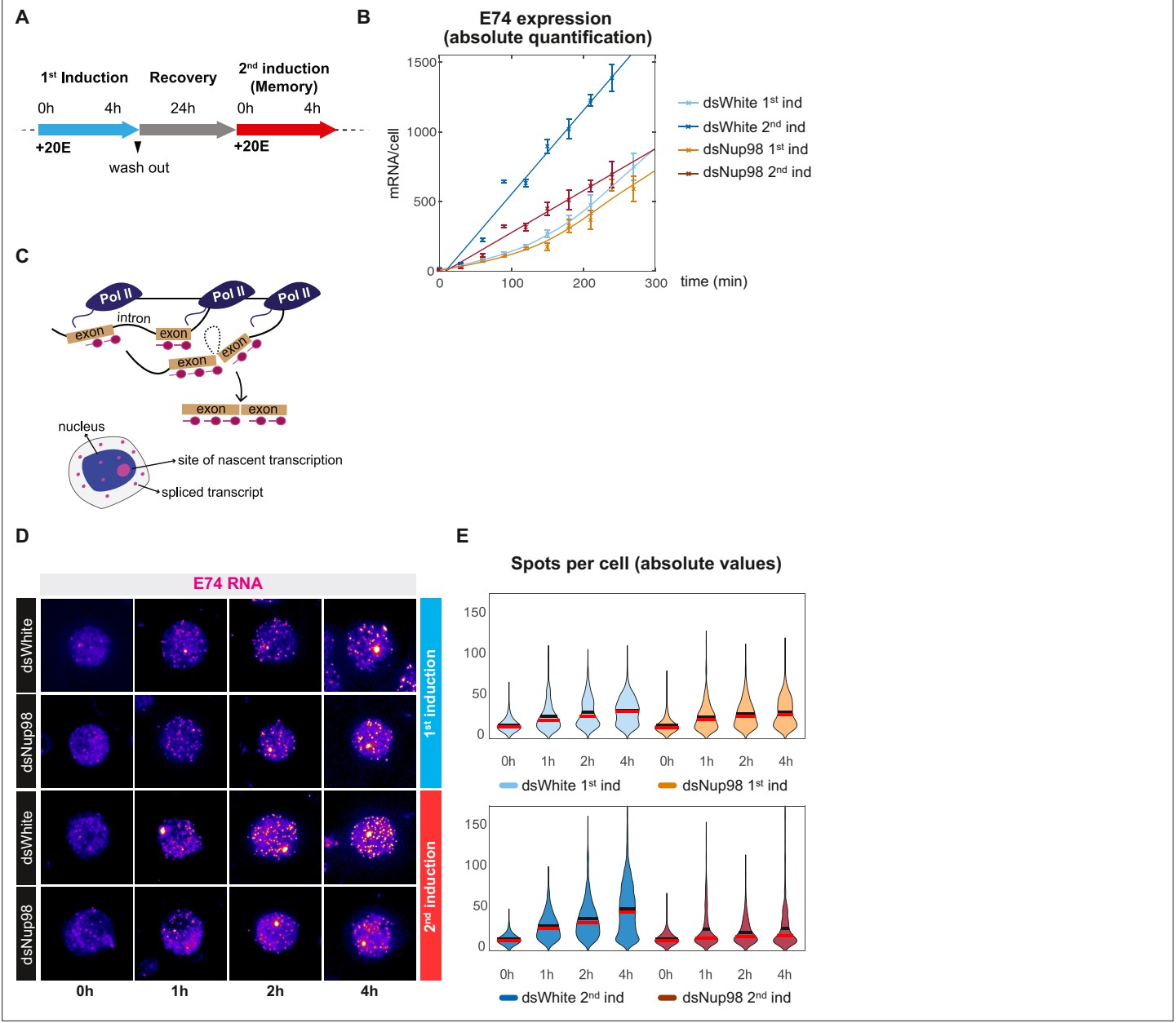

**Figure 1.** Absolute quantification of *E74* induction and transcriptional memory. (**A**) Overview of ecdysone/20E treatment. Cells were treated with 5 µM 20E, then washed with fresh media and recovered for 24 hr. Memory response was assessed by incubating cells with 5 µM 20E after recovery period. (**B**). Absolute number of *E74* mRNAs per cell as a function of time after the addition of 20E during either the first or second induction in control (dsWhite) and Nup98 knockdown (dsNup98) S2 cells. Samples were collected every 30 min. Error bars represent standard deviation of the mean of three experiments. (**C**). Schematic of smFISH labeling of sites of nascent transcription and spliced transcripts in either the nucleus or cytoplasm. (**D**). Representative images of *E74* smFISH labeling in single cells during the first and second induction, displayed with the ImageJ 'red fire' look-up table. (**E**). Violin plots of *E74* puncta per cell. Mean and median indicated by black and red horizontal lines.

The online version of this article includes the following figure supplement(s) for figure 1:

**Figure supplement 1.** Absolute quantification of *E74* expression.

**Figure supplement 2.** Analysis of *E74* smFISH resolution.

*E74* expression is uniform between cells, and to gain insight into the nature of hormone-induced memory response, we employed smFISH to measure *E74* expression in single cells. *E74* mRNAs were labeled with a set of 67 probes complimentary to exon sequences (***Figure 1C*** and ***Supplementary file 1***) and imaged by confocal microscopy. smFISH reveals two classes of labeled objects: relatively dim,

diffraction-limited puncta enriched in the cytoplasm representing mature mRNAs, and one or more bright nuclear-localized puncta corresponding to sites of nascent transcription (*Figure 1C–D*). To estimate *E74* mRNA levels, we counted the number of mRNA puncta at 0, 1, 2, and 4 hr post-induction and observed that the average number of puncta per cell accumulated with dynamics that mirrored those obtained by absolute qPCR (*Figure 1D–E* and *Figure 1—figure supplement 1C-D*). mRNA content of puncta also increased in a manner consistent with qPCR (*Figure 1—figure supplement 1E-F*). As expected, during the first induction, dsWhite-treated and Nup98-depleted cells accumulated puncta at similar rates, while during the second induction, control cells accumulated mRNA puncta more rapidly and to higher levels than Nup98-depleted cells or cells undergoing the initial induction (*Figure 1E*). Such accumulation of mRNA puncta was not seen in smFISH of control probes against genes that do not respond to ecdysone (*Shlyueva et al., 2014*), which exhibited similar counts before and after ecdysone addition (*Figure 1—figure supplement 2A-C*). The validity of these smFISH counts is also supported by our demonstration that the detected objects are diffraction limited (*Figure 1—figure supplement 2D-F*, *Petkova et al., 2014*; *Little et al., 2015*). Importantly, we observed that *E74* expression is heterogeneous between cells, with mRNA amounts exhibiting broad distribution at all times and under all induction conditions. Even under the most highly expressed condition at 4 hr after the second induction, 20% of cells possess 20 puncta or fewer, whereas <5% contain 100 or greater. These smFISH results validate our quantitative RT-qPCR data, but do not support the simple model of *E74* activation or production occurring at a uniform rate. The observed heterogeneous mRNA levels across cells in all conditions also suggest that the role of Nup98 is to increase average expression levels without increasing the uniformity of the hormone response during the second induction. Consistently, analysis of transcriptional noise in the four conditions reveals that Nup98 knockdown does not alter variability in mRNA counts between cells (*Figure 1—figure supplement 2G*).

## Nup98 modulates *E74* mRNA production without affecting export or degradation

To further explore the role of Nup98 in the memory response, we undertook additional quantitative measurements of *E74* mRNA dynamics. Our preceding results (*Figure 1*) confirmed a Nup98-dependent increase in the rate of *E74* mRNA accumulation during the second induction. Generally, the rate of mRNA accumulation depends on both mRNA production and degradation. Degradation of mRNA may be affected by the process of mRNA export, which interfaces with NPCs (*Rodríguez-Navarro and Hurt, 2011*; *Tutucci and Stutz, 2011*). Certain NPC components, including Nup98, have been previously implicated in mRNA export through interactions with the mRNA export machinery (*Blevins et al., 2003*; *Chakraborty et al., 2006*; *Powers et al., 1997*), and mRNA export could in turn affect mRNA stability either directly or indirectly. Since prior work has not addressed whether Nup98 affects *E74* mRNA levels through these processes, we first asked whether Nup98 knockdown induces a change in *E74* mRNA trafficking out of the nucleus or in the rate of degradation of *E74*.

To answer these questions, we examined the smFISH images for evidence of altered nucleocytoplasmic transport by assessing the fraction of mRNA puncta that are found outside of the Hoechst-based mask out of the total amount of mRNA puncta per cell (*Figure 2—figure supplement 1A*). We found that this fraction does not change in control/dsWhite versus dsNup98 cells significantly in either induced condition, with >85% of mRNAs found in the cytoplasm across conditions (*Figure 2A*). Next, to determine whether *E74* mRNA stability is altered upon Nup98 knockdown, we monitored *E74* mRNA levels following transcriptional arrest. We inhibited transcriptional elongation by treating cells with flavopiridol (FP), a potent inhibitor of p-TEFb, which phosphorylates Ser2 of the C-terminal domain of RNA Pol II (*Chao and Price, 2001*). *E74* mRNA synthesis was stimulated with 20E for 4 hr during the first or the second inductions, after which the hormone was washed out and transcription blocked with FP. As the mRNA levels of *E74* declined, we monitored them for 240 min, with time points collected every 30 min, during the first or the second inductions in control or Nup98-depleted conditions (*Figure 2B*). We observed that the degradation rates were largely unchanged between Nup98-depleted and control cells during either induction, with no significant difference in *E74* mRNA lifetimes, which were determined by fitting to a first-order reaction (*Figure 2B* and *Figure 2—figure supplement 1B*). We concluded that neither mRNA export nor mRNA degradation are major contributors to the phenotype of Nup98 in transcriptional memory. Instead, our data suggests that the role of Nup98 in transcriptional memory is mediated through regulation of transcription, which is supported

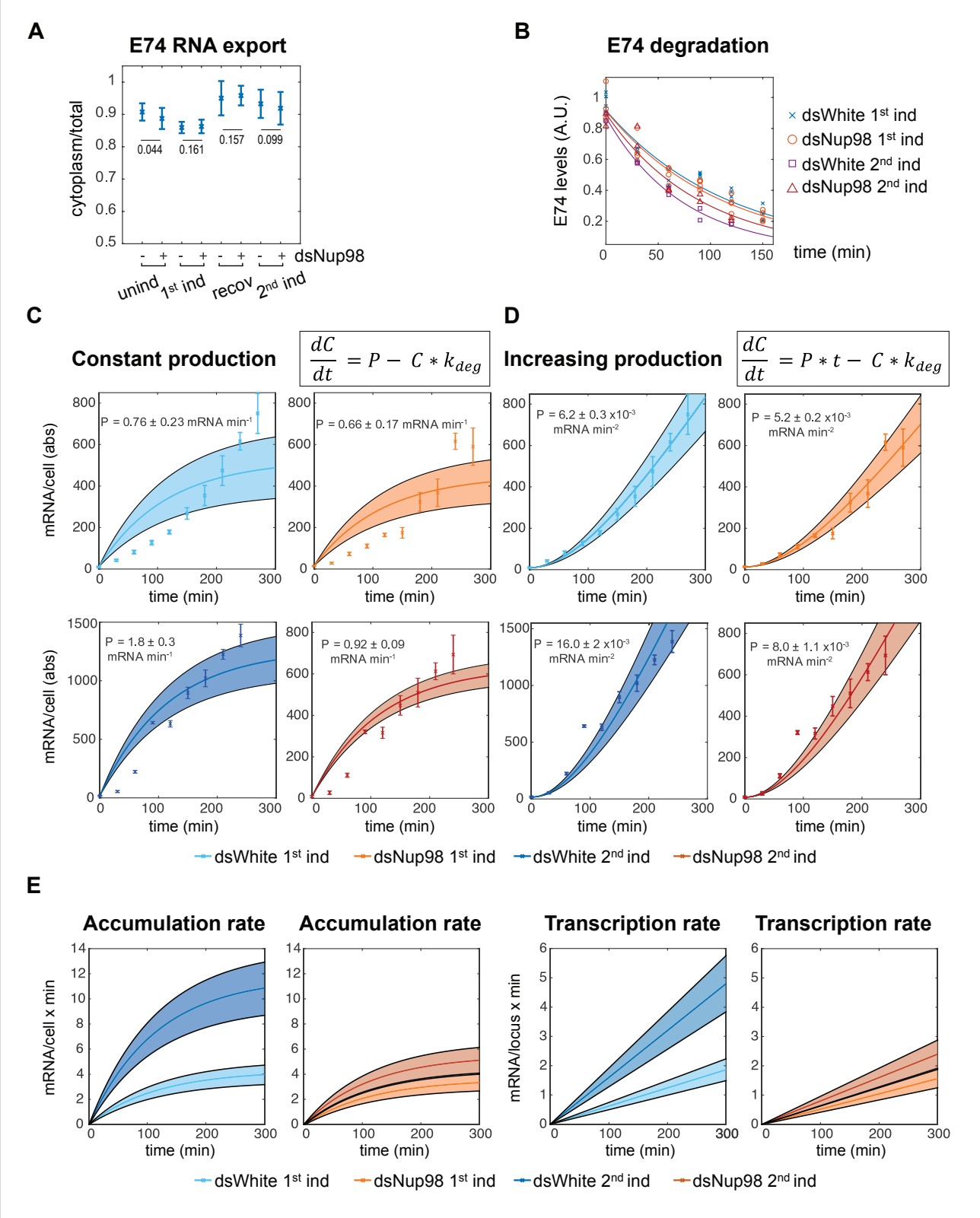

**Figure 2.** Altering Nup98 levels modulates transcription, not mRNA export or degradation. (**A**) Fraction of *E74* mRNA found in cytoplasm, as assessed by smFISH, with Nup98 knockdown (+) or control knockdown (-) prior to the first induction (unind), after 4 hr (1st ind), 24 hr after hormone removal (recov), and 4 hr after 2nd induction. Error bars represent standard deviation of the mean. Welch's t-test was used to calculate p-values. (**B**). Degradation rates of *E74* mRNA assessed by qPCR. Cells from 1st induction or 2nd induction (as depicted in ***Figure 1A***) were washed and collected every 30 min

*Figure 2 continued on next page*

*Figure 2 continued*

after the addition of 1 μM flavopiridol at zero minutes. Experiment was repeated three times and mRNA lifetimes were determined by fit to exponential curves. p-Values, determined by Welch's t-test, have values > 0.2 for all pairs of fits. (**C–D**). *E74* transcription rates increase during 20E exposure. Data points with error bars obtained by qPCR as shown in *Figure 1B*. Shaded bands indicate 95% confidence intervals for the fit rates P for the models indicated. Top row: first induction; second row: second induction. (**C**) Best fit of data to model of constant transcription rate *P*. (**D**): Best fit of data to model of time-dependent increasing *E74* transcription rate. (**E**). Comparison of the rates of mRNA accumulation (left) and underlying transcription rates (right) between first and second induction in control and dsNup98 cells.

The online version of this article includes the following figure supplement(s) for figure 2:

**Figure supplement 1.** Examination of mRNA detection, export, and lifetime.

by the previously identified binding of Nup98 to the promoters and enhancers of ecdysone-inducible genes (*Pascual-Garcia et al., 2017*).

Our combined knowledge of the degradation rate with the accumulation dynamics in absolute units allowed us to test quantitative models describing the effect of Nup98 on transcription. In the simplest model, Nup98 would act to boost the average transcription rate from a low constant mRNA production rate during the first induction to a large constant rate during the second induction. To determine whether a model of constant production correctly describes the data, we found the best-fitting values for the production rates under each induction and knockdown combination (*Figure 2C*). In this straight forward model, the change in mRNA concentration per time equals mRNA production rate P minus the degradation rate, which was determined experimentally (*Figure 2B*). The estimate of the degradation rate permitted us to obtain best-fit values for P (*Figure 2C*). As expected, the value for production rate for the second induction in control cells was higher than during the first induction by a factor of 2.4 x, whereas the production rates are similar in first induction control and Nup98 knockdown conditions under both exposures. However, constant production rates provide a poor description of the data (*Figure 2C* and *Figure 2—figure supplement 1C*). In all cases, the constant production model predicts the fastest accumulation during the first 2 hr of expression, with gradual attenuation of the accumulation as transcript levels reach steady-state. The data instead exhibits a gradual and slow increase in the rate of accumulation during the majority of interval for hormone exposure. This is most obvious for the first induction in both control and Nup98 knockdown cells. The results indicate that the simplest explanation for Nup98 activity, boosting a constant production rate, is incompatible with observation.

Instead of constant production rate, the data suggested that the production of *E74* mRNA increases with time. We therefore fit a model in which production of new mRNAs increases linearly with time, and found that this model provided an excellent fit (*Figure 2D* and *Figure 2—figure supplement 1C*). We observed that the production rates increase at similar rates in three scenarios: first induction in control, and both inductions in Nup98 knockdown conditions. In contrast, during the second induction in control cells, the production rate increases at more than twice the rate of the first induction. This can be clearly visualized by taking the derivative of the fit accumulation curves to reveal the accumulation rate, that is the change in the average number of mRNAs per cell as a function of time (*Figure 2E*). The accumulation rates are similar in control first induction and dsNup98 first and second induction, but become increasingly fast in control second induction. The accumulation rates begin to stabilize at around 4 mRNAs per minute during the first induction but reach greater than 10 mRNAs per minute during the second induction in control cells (*Figure 2E*, left). During hormone exposure, the transcription rate climbs continuously, reaching only about 2 mRNAs per minute per locus during the first induction but climbing to 5 mRNA per minute per locus during the second induction in control cells (*Figure 2E*, right). By the end of the fourth hour of the second exposure, this corresponds to an average production rate per locus similar to that observed for the most highly expressed genes in *Drosophila* embryos (*Little et al., 2013*; *Zoller et al., 2018*). Overall, these results show that Nup98 changes transcription of *E74* without affecting mRNA trafficking or stability. Moreover, normal levels of Nup98 are required not simply to promote a higher level of transcription, but to stimulate a continuous increase in transcription rates upon additional hormone exposure.

## A two-state model for the mechanism of Nup98-dependent transcription rate increase

We next sought to explore models describing the Nup98-dependent increase in the rate of mRNA production. We began with a two-state model in which loci switch between active and inactive states (*Figure 3A–B*). During the active state, new RNA Pol II molecules enter productive elongation at a rate given by $k_{Pol}$. For simplicity, we assumed that while hormone is present, the rate switching into the active state $k_A$ is much greater than the rate of switching into the inactive state $k_{-A}$. This ensures that once active, loci are in essence irreversibly committed to the active state as long as the hormone is present. These assumptions are reasonable given the large but slow increase in transcript levels observed over the duration of the hormone exposure. In this model, the average production rate per locus is given by the loading rate $k_{Pol}$ multiplied by the fraction of active loci (*Figure 3B*).

There are then two basic ways in which the mRNA production rate can increase over time in a two-state model. The most straightforward is by recruitment of loci into the transcriptionally active state, while the loading rate $k_{Pol}$ is held constant (*Figure 3A*, Model 1 and *Figure 3B*). In this scenario, what would change over time is the fraction of cells in a transcriptionally active state (*Figure 3A*, Model 1). Once a locus enters an active state, mRNA production would occur at a constant rate. The increased transcriptional output during re-induction would thus stem from a higher fraction of cells being active in post-memory conditions. This scenario requires a $k_A$ that is small such that any given locus is unlikely to enter the active state in any given minute. The gradual increase in the fraction of active loci would thus be proportional to the continuous increase in the average transcription rate (*Figure 2E*, right). Under Model 1, the role of Nup98 is to increase $k_A$ such that a larger fraction of loci become active sooner during the second induction.

We fit this model to the qPCR data. For each of the four scenarios (control/dsWhite or dsNup98, first or second induction), we found values and confidence intervals that described the accumulation curves (*Figure 3—figure supplement 1A*). Fitting each scenario independently, we found that $k_{Pol}$ was roughly constant at 2.0±0.2 RNA Pol II per minute for all scenarios (average and standard deviation calculated across all four scenarios) (*Figure 3—figure supplement 1B*). Moreover, fit values of $k_A$ are very similar for the three low expressing scenarios, dsWhite first induction and both inductions in dsNup98, at an average of 5.1±1.6 × $10^{-3}$ per minute (*Figure 3—figure supplement 1B, E*). In contrast, $k_A$ for the second induction in control is 33±15 × $10^{-3}$ per minute, a sixfold increase (*Figure 3—figure supplement 1B, E*). In this model, Nup98 is required for this dramatic upregulation in rate of entry into the active state during the second induction.

Although Model 1 can explain increasing transcription rate at the population level, we can also envision an alternative scenario, where all cells enter an active state rapidly upon hormone stimulation, and what increases over time is the transcriptional rate at individual loci (*Figure 3A*, Model 2). The two-state model provides an additional means of regulation, in which $k_{Pol}$ is not constant but instead increases with time (*Figure 3A*, Model 2, and *Figure 3B*). In the most extreme version of this scenario, $k_A$ is very large such that all loci switch into the active state immediately upon hormone exposure. This would be consistent with the known rapid rate of nuclear import of hormone receptors, on the order of minutes (*Nieva et al., 2007*). In this scenario, the increasing transcription rate would result from increasing $k_{Pol}$, and the increased rate of mRNA production during second induction would be explained by a faster increase in transcriptional rate at individual loci. Under Model 2, the role of Nup98 would be to ensure a more rapid increase in $k_{Pol}$ upon repeated hormone treatment.

To determine whether this scenario could explain our observations, we fit our qPCR data to a model in which (1) $k_A$ is very fast (1000 per minute) to ensure that nearly all loci become active within a minute of hormone exposure, and (2) the number of attempts that RNA Pol II makes to engage in transcription increases linearly with time. The fraction of attempts that are successful is limited by process(es) that prohibit Pol II from loading at an indeterminately high density, thereby imposing a minimum interval between transcribing Pol IIs. In implementing the model, the minimum interval is determined by fitting the qPCR data with a constant elongation rate of 1500 nt/min (*Izban and Luse, 1992*, *Ardehali and Lis, 2009*; *Buckley et al., 2014*; *Yao et al., 2007*) and introducing a free parameter representing the minimum interval in units of nucleotides. The model is agnostic with regard to the molecular mechanism(s) underlying either the rate increase or the minimum Pol II interval. With these constraints we were able to fit the qPCR data from each scenario to a model of increasing RNA Pol II attempt rate that described the accumulation trajectories (*Figure 3—figure supplement 1C-D*).

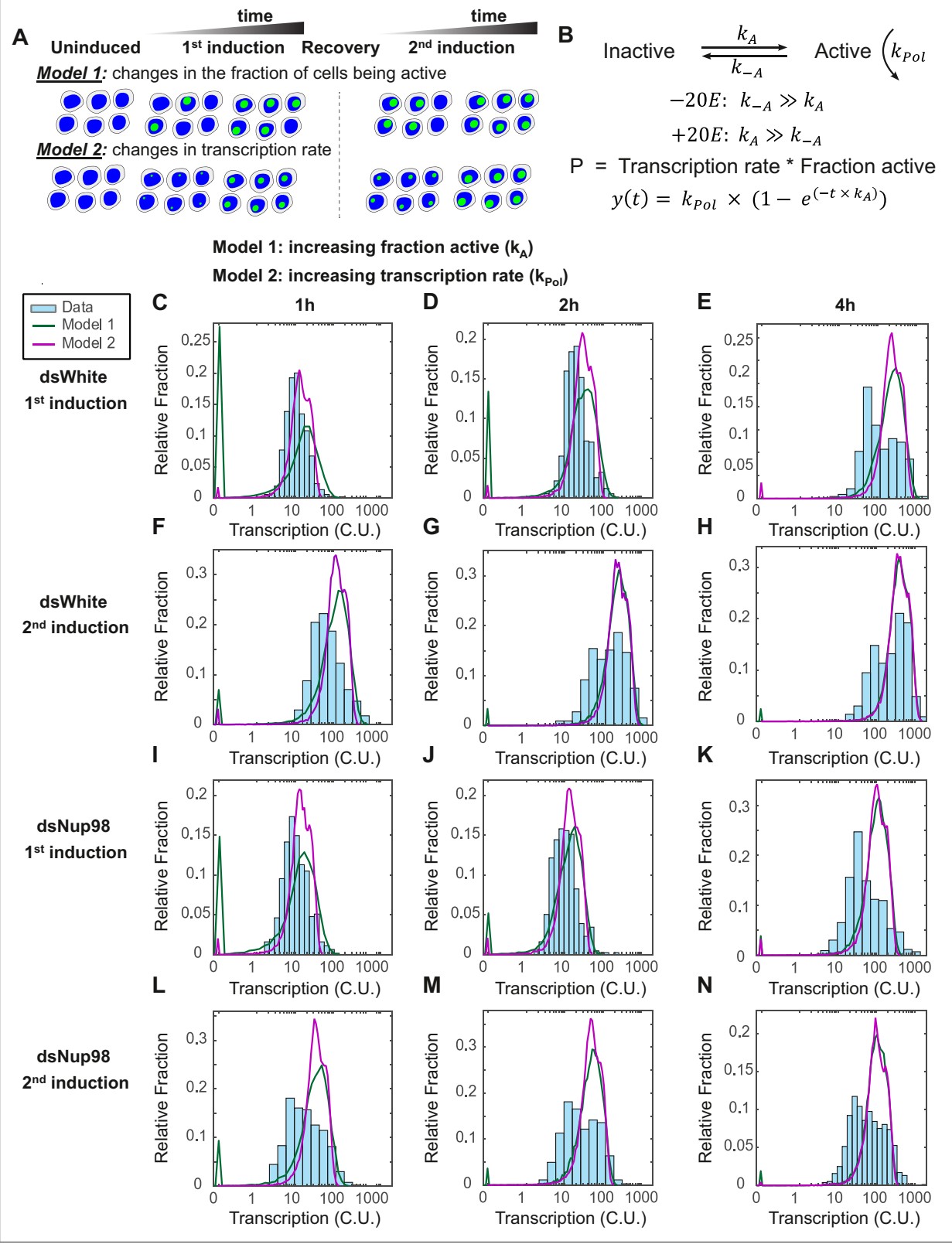

**Figure 3.** The two-state promoter model does not describe single-cell measurements of transcription. (**A**) A model utilizing two promoter states (active and inactive) contains two mechanisms that can account for the increase in the transcription rates observed by qPCR. Model 1: loci are slowly recruited into a transcriptionally active state upon first induction and more rapidly upon second induction. Once active, loci produce new mRNAs at a constant rate equivalent to the rate at which new RNA Pol II molecules enter productive elongation. Model 2: all cells are rapidly recruited into an

*Figure 3 continued on next page*

*Figure 3 continued*

active state, and the rate of transcription increases over time, slowly upon first induction and quickly upon second induction. (**B**). Model describing the rate of recruitment into the active state $k_A$, into the inactive state $k_{-A}$, and the rate of recruiting RNA Pol II molecules into productive elongation $k_{Pol}$. The presence of 20E permits $k_A$ to be larger than $k_{-A}$. The production rate as a function of time is given by the RNA Pol II recruitment rate multiplied by the fraction of active loci. In Model 1, $k_A$ increases and $k_{Pol}$ is constant during 20E exposure, whereas in Model 2, $k_{Pol}$ increases and $k_A$ is constant. (**C–N**). Histograms (cyan) show distribution of measured total instantaneous transcriptional activity in normalized units (C.U.), obtained from smFISH of *E74* as shown in *Figure 1*. Lines represent predicted values generated by simulation using best-fitting parameters for Model 1 (green) and Model 2 (magenta) under conditions of control (**C–H**) or Nup98 knockdown conditions (**I–N**) during the first (**C,F,I,L**), second (**D,G,J,M**), or fourth (**E,H,K,N**) hour of the first (**C–E, I–K**) or second (**F–H, L–N**) inductions.

The online version of this article includes the following figure supplement(s) for figure 3:

**Figure supplement 1.** Modeling promoter state switching and increasing transcription rates using population-averaged qPCR data.

**Figure supplement 2.** Estimated population distributions of cells with 0–4 nascent transcription sites as a function of time.

As before, three conditions (dsWhite first and dsNup98 first and second inductions) have similar acceleration constants, with an average of $11.7\pm2.2 \times 10^{-3}$ Pol II min$^{-2}$ (*Figure 3—figure supplement 1E*). In contrast, control second induction has an acceleration constant that is about six-fold larger than the first at $65.7 \times 10^{-3}$ Pol II min$^{-2}$ (*Figure 3—figure supplement 1D-E*). In this model, the role of Nup98 is to increase the acceleration of the attempt rate $k_{Pol}$.

Overall, both versions of the two-state model perform equally well at describing the qPCR data (*Figure 3—figure supplement 1F*). Additionally, the two versions are not mutually exclusive; our qPCR data would be equally well fit by many possible combinations of values for $k_A$ and $k_{Pol}$ acceleration constants. To discern between these possibilities requires single-cell measurements of transcription provided by smFISH.

## Single-cell analysis does not support the two-state model

To discern between possible versions of the model outlined above, we proceeded to determine whether they could account for the distribution of transcriptional activity observed in our smFISH analysis of *E74* (*Figure 1*). We determined the instantaneous transcriptional activity in individual cells by summing the fluorescence intensity of all nascent transcription sites in each cell and normalizing to the intensity of single mRNAs (also see Materials and methods). The resulting values are the absolute amount of nascent RNA present at transcribing loci in units of the equivalent number of mature mRNAs (*Little et al., 2013*; *Zoller et al., 2018*). As mRNAs tend to be localized to the cytoplasm, the unit intensity is referred to as the 'cytoplasmic unit' intensity (C.U.). The number of C.U.s associated with each transcribing site depends on the number of probe binding sites present, and thus on how many RNA Pol II molecules are present and how much of the transcript has been synthesized by each Pol II. The normalization is possible because our probes target only exon sequences and thus the fluorescence intensity is not affected by mRNA splicing or the presence of introns (*Figure 1C*).

To determine if either version could account for the actual transcriptional behavior in ecdysone-induced responses, we used the values for $k_A$ and $k_{Pol}$ from the fit to qPCR to simulate the distribution of C.U.s for both models at three time points (1, 2, and 4 hr post-induction) under all four induction conditions. Monte Carlo simulations were used to generate RNA Pol II positions for 100,000 cells under each condition and time point. The RNA Pol II positions were used in combination with the probe positions along the transcribed RNA to obtain a sum of probe binding sites present in the simulated nascent mRNA. This sum was then normalized to the number of probes present in a single mRNA to generate simulated distributions in units of C.U.s. Histograms of measured and simulated C.U. distributions were then compared (*Figure 3C–N*; see Materials and methods).

We examined the distribution of transcriptional activity across hormone treatments and time points (cyan bars, *Figure 3C–N*). Notable features in the observed distribution of transcriptional activity include a gradual shift in the activity levels as a function of time, as expected from the RT-qPCR data, meaning that an increasing fraction of cells shift to transcribing at higher amounts. This shift occurs more rapidly for the control/dsWhite second induction (*Figure 3F–H*) than for other conditions. The observed distribution begins to resemble a bimodal distribution in control cells at 4 hr (*Figure 3H*), demonstrating that a subset of nuclei have begun transcribing *E74* at rates much faster than earlier in the exposure, as measured in C.U.s. This is again consistent with the prediction from qPCR that the transcription rates in general are increasing. Notably, this bimodal behavior is less evident in

dsNup98 conditions (compare *Figure 3L–N*), where the highly transcribing fraction of cells is much less pronounced at 4 hr of second induction (*Figure 3N*). This suggests that Nup98 could be involved in a shift from low to high expression rates. Additionally, the change in the estimated fractions of cells with 0–4 active loci over time appears similar among the four conditions (*Figure 3—figure supplement 2A-E*, see Materials and methods for description of how active transcription sites were designated), supporting the notion that the rate of locus-associated transcriptional activity is the point of regulation.

We compared the distribution of the data to those predicted by each of the two-state models. Overall, the simulated distributions from either of the two two-state models provide a poor fit to data (*Figure 3C–N*, compare simulated green and violet curves to cyan bars of observed data, and *Figure 4—figure supplement 1D*). Model 1, with changing $k_A$ and constant $k_{Pol}$, consistently overestimates the fraction of cells with small numbers of transcripts at early times (green curves, *Figure 3C,I*, notice the early green 'spike'). Model 2, with rapid $k_A$ and increasing $k_{Pol}$, suffers less from an overestimate of inactive cells. However, both models predict a narrowly distributed peak of highly active cells at 4-hr time point in all four conditions. In contrast, measurements reveal broadly distributed expression levels, with many cells showing less activity than predicted (*Figure 3E, H, K and N*). Furthermore, hybrid models combining both more rapid $k_A$ with more slowly increasing $k_{Pol}$ suffer from both shortcomings in combination (data not shown). This stems from the failure of the two-state model to account for the large fraction of cells containing mRNAs in the first hour combined with an overestimate of narrowly distributed but highly active cells at late times. From the simulation results, we concluded that no two-state model correctly captures the transcription dynamics of the hormone response in either the first or second induction.

## A four-state 'Memory Switch' model explains single-cell data of transcriptional memory

Given the distribution of transcriptional states we observed in our single-cell smFISH analysis (*Figure 3*), we reasoned that loci may express *E74* at two different rates, such that loci are either in the low- or high-expressing state in the presence of ecdysone. Both the low- and high-expressing rates would be larger than the very slow basal expression rate observed in the absence of ecdysone. The slow emergence of the high-expressing state during exposure to ecdysone would be consistent with the gradual increase in transcription rate we have observed (*Figure 2E*). Moreover, the rapid accumulation of transcripts upon hormone re-exposure suggests a mechanism for memory: loci that previously converted into the high-expressing state bypass the low-expressing state upon hormone re-exposure, and immediately enter the high-expressing state. We termed such converted state the induced memory (IM) state, whereas loci that have not converted remain in a default low-expressing state, which we called induced default (ID) state (*Figure 4A*). Importantly, the model also suggests that the high expression observed during the second induction results from the continuous accumulation of loci into the memory state even after the withdrawal of hormone. In this case, the role of Nup98 may be both to ensure that converted loci retain information about their conversion state and to enact the continuous conversion of loci after hormone withdrawal (*Figure 4A*, right). This would explain the strong resemblance of the first induction in controls cells to both inductions when Nup98 is depleted: in the absence of hormone, all loci reconvert to the default, non-memory state without Nup98 activity.

Based on this reasoning, we constructed a 'memory switch' model containing four states: two un-induced states in the absence of hormone, termed Un-induced Default (UD) and Un-induced Memory (UM) (both characterized by very low basal levels of transcription), in addition to the two induced states, IM and ID (*Figure 4A*). In this model, the basal level of expression in the absence of ecdysone is described by the transcription rate constant $k_{polB}$, while hormone exposure has two independent effects. First, similarly to the two-state model, hormone switches non-expressing loci into an expressing state at rate $k_A$. As before, hormone ensures that the conversion to expression is irreversible as long as hormone is present. Second, hormone treatment converts loci from the default state (either UD or ID) into the memory state (either UM or IM). This conversion occurs at rate $k_C$ and is necessarily slow. The two induced states each have an associated rate of RNA Pol II loading, low $k_{PolL}$ for ID and high $k_{PolH}$ for IM. To account for the immediate entry of loci into the high expressing IM state upon second induction, the memory state is established irreversibly, with $k_C$ much greater than $k_{-C}$ in normal conditions. Moreover, the model implies that once the hormone is applied, the conversion of

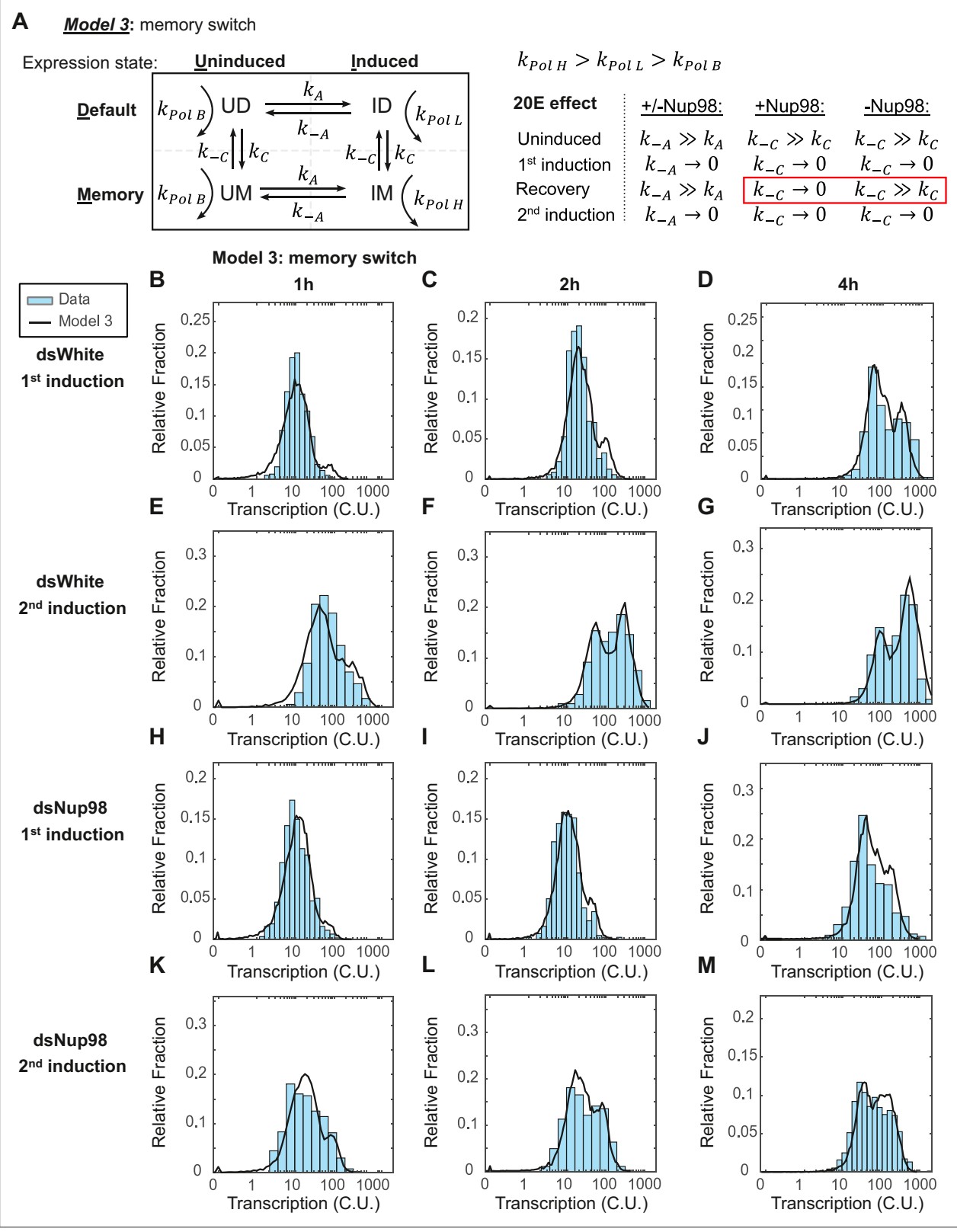

**Figure 4.** The memory switch model describes the distribution of transcriptional activity. (**A**) Promoters can occupy one of four states: Uninduced and Induced (**U and I**), related to current presence of hormone; and Default and Memory (**D and M**), associated with prior hormone exposure. The Uninduced state is associated with a basal RNA Pol II rate $k_{PolB}$, whereas the Induced state shows two independent RNA Pol II rates $k_{PolL}$ and $k_{PolH}$ associated with Default and Memory states, respectively. 20E has two roles: one, to activate transcription by ensuring $k_A >> k_{-A}$, as in earlier models; and

*Figure 4 continued on next page*

*Figure 4 continued*

two, to increase the rate of conversion from Default to Memory by ensuring $k_C \gg k_{-C}$. The role of Nup98 is to maintain $k_C \gg k_{-C}$ upon withdrawal of 20E. (**B–M**). Histograms (cyan) show distribution of measured total instantaneous transcriptional activity in normalized units (C.U.), obtained from smFISH of *E74* as shown in *Figure 1*. Lines represent predicted values generated by simulation using best-fitting parameters under the memory switch model.

The online version of this article includes the following figure supplement(s) for figure 4:

**Figure supplement 1.** Fitting qPCR data to a four-state model of promoter memory.

loci to the memory state continues even after the hormone has been withdrawn, so that after the 24 hr recovery period in our hormone treatment regimen, most loci are found in the memory state. Under this model, the primary role of Nup98 would be to maintain a high $k_C$, or the probability that cells remain in the memory or high-expressing state, such that depletion of Nup98 abrogates the maintenance of the memory state upon removal of hormone. In the absence of Nup98, $k_{-C}$ becomes greater than $k_C$, and loci do not remember their prior exposure once hormone is withdrawn (*Figure 4A*, right), which explains why the second induction strongly resembles the first in Nup98-depleted cells.

If this model is correct, then a single set of values for the parameters $k_C$, $k_A$, $k_{PolL}$, and $k_{PolH}$ should describe both the qPCR and smFISH results from all four conditions. To test this, we started by fitting the model to the qPCR data for each individual condition. We used the functions describing the average transcription rate over time (*Figure 2E*, right) as a constraint to narrow the range of allowable parameter values capable of recapitulating the functions. We then performed Monte Carlo simulations, searching a wide range of parameter values to find sets that reproduced the smFISH data. The resulting values captured the rise in the transcription rate and the resulting accumulation trajectories observed by qPCR under all four conditions (*Figure 4—figure supplement 1A-C*). As required by the model, the fraction of cells in either expressing state increased in a similar manner over time between all conditions (*Figure 4—figure supplement 1B*). This conversion into either expressing state ID or IM occurs at the same rate (on average, $k_A = 17 +/- 1 \times 10^{-3}$ per min, *Figure 4—figure supplement 1C*; see also *Figure 4A*). In contrast, the conversion into either memory state UM or IM is very slow ($k_C = 0.8 +/- 0.2 \times 10^{-3}$ per min) (*Figure 4—figure supplement 1C*; see also *Figure 4A*), meaning the characteristic decay time out of the default state is very long ( > 1200 min or about 20 hr). This means that the more time that passes after the first induction, the larger the fraction of loci that convert to the memory state, even after the withdrawal of hormone. In this manner, after the 24 hr recovery, the majority of loci (74%) are in the memory state (*Figure 4—figure supplement 1E*), ready to enter the high-expressing state immediately upon re-exposure.

To verify the goodness of the model, we again performed Monte Carlo simulations and compared the resulting simulated distributions to smFISH data for *E74*. The simulation closely matches observation (*Figure 4B–M*) and provides a vastly improved fit relative to either of the two-state models (*Figure 4—figure supplement 1D*). The model thus successfully captures the emergence of the most highly-expressing cells during the first induction as a consequence of the slow conversion from non-memory to memory states. Notably, the rate of RNA Pol II loading is 15 times greater for the high-expressing IM state ($k_{polH}$ 4.5 versus $k_{polL}$ 0.3 RNA Pol II per minute, *Figure 4—figure supplement 1C*) and approaches the maximum observed value found in *Drosophila* embryos (*Little et al., 2013*). The vastly increased expression rate during the second induction again requires Nup98 function; upon depletion, loci fail to re-enter the high-expressing state upon re-exposure and are found in the default state upon re-induction (*Figure 4—figure supplement 1E*).

In summary, the four-state memory model encapsulates all the essential features of our observations. Our analysis puts forth a mathematical model that accurately describes the transcriptional memory behavior of ecdysone-inducible genes. The analysis predicts that in addition to activating default transcription, hormone exposure leads to a response event that drives a switch between non-memory and memory states.

## The establishment of transcriptional memory is independent of transcriptional activity during the initial induction

The memory switch model, described above, reveals novel mechanistic insights into how transcriptional memory is established and maintained, and allows us to make predictions that can be tested experimentally. Thus, to further validate our model, we proceeded to test two of its main predictions.

The most striking and surprising prediction of the model is the independence of $k_C$ (the rate of conversion from the default state to the memory state) from $k_A$ and $k_{Pol}$, which are rate constants that describe the transcriptional process itself. Otherwise stated, the model suggests that the ability of cells to establish the transcriptional memory response is independent from the process or the extent of transcription itself. The model predicts that at individual loci, addition of ecdysone sets off the conversion into the high-expressing memory state, but the parameters governing the rate of the conversion should be independent from the transcription that takes place during the first induction.

We tested this prediction by two different approaches. First, we varied the amount of transcription by varying the length of exposure to 20E during the initial induction. We assessed the transcriptional memory response of the *E74* gene via RT-qPCR, using diminishing initial incubation times of 20E ranging from 4 hr to a minimum of 10 min, after which cells were recovered for 24 hr and re-induced as usual (*Figure 5A*). In agreement with the model, we observed that cells responded with a similarly robust increase of transcription during the second induction after all initial hormone incubation times. No large changes were found in the *E74* transcriptional memory response between cells that were induced for 10 min versus 4 hr (*Figure 5A*), demonstrating that the memory state is established independently of the length of 20E incubation times and thus of the length of time these loci were engaged in active transcription.

Second, to address this prediction, we utilized the transcriptional inhibitor FP to prevent transcriptional elongation altogether. Cells were treated with FP for 30 min before as well as during 20E induction. We monitored *E74* expression and observed no transcriptional activity in cells treated with FP during ecdysone induction, revealing the efficacy of FP blockage (*Figure 5B*). Cells were then washed, and we measured the transcriptional memory response after the usual 24 hr of recovery. Strikingly, we observed that blocking transcriptional elongation during the first induction does not substantially affect the ability of the cells to generate a robust memory response, indicating that the transition into the memory state is independent of mRNA production during the first induction (*Figure 5B*). Levels of *E74* were normalized to transcript levels of the housekeeping gene *rp49*, expression of which has been shown to be unaffected by ecdysone exposure (*Shlyueva et al., 2014*; *Figure 5—figure supplement 1A*). In addition to FP, we tested another transcriptional inhibitor Triptolide (TPL), which is known to block transcription at an earlier, initiation-associated point by inhibiting TFIIH (*Titov et al., 2011*). Treatment with TPL during initial induction similarly resulted in the majority of transcriptional memory being preserved (*Figure 5B*), although at somewhat lower levels than FP, suggesting that transcriptional initiation may partially contribute to memory, but transcription itself does not. We further validated our RT-qPCR findings by smFISH analysis, which likewise yielded similar counts of *E74* mRNA spots and transcriptional output during second induction, in control or after treatment with either FP or TPL in the initial induction (*Figure 5C* and *Figure 5—figure supplement 1B-C*). Importantly, we obtained very similar conclusions with two other ecdysone-inducible genes *E23* and *E75*, which we previously found to depend on Nup98 for transcriptional memory (*Pascual-Garcia et al., 2017*) and which also retained most of their transcriptional memory after transcriptional inhibition (*Figure 5—figure supplement 1D-E*). Taken together, our experimental data support the prediction proposed by our modeling studies and reveal a previously unreported feature of transcriptional memory: that its establishment is independent from the extent of transcriptional activity during initial stimulation.

The second prediction of the model that we aimed to test was the slow conversion from non-memory to memory state, on the order of 20 hr (for a conversion of 62% of loci) based on the obtained parameters. We tested this timescale by shortening the recovery times after ecdysone induction. Cells were induced with 20E for 1 hr and recovered for six or 24 before testing transcriptional memory responses. The model anticipates that cells that have been recovered for 6 hr should have a lowered memory response than cells that have been recovered for 24 hr. Consistently with this prediction, we found a reduced memory response for cells recovered for 6 hr (*Figure 5D*), underpinning the notion that transition into the memory state relies on mechanisms that (1) do not require the continuous presence of ecdysone and (2) have a relatively long timescale.

In order to compare the differences in transcriptional memory responses caused by distinct factors or treatments, we represented the observed differences in transcriptional memory responses as Calibrated Memory index (CMI), derived from the ratios of slopes in mRNA accumulation during second inductions and calibrated to control and Nup98-disrupted responses, as measured by RT-qPCR for *E74*, *E23*, or *E75* (*Figure 5E*; see also Materials and methods). Comparing CMI among different

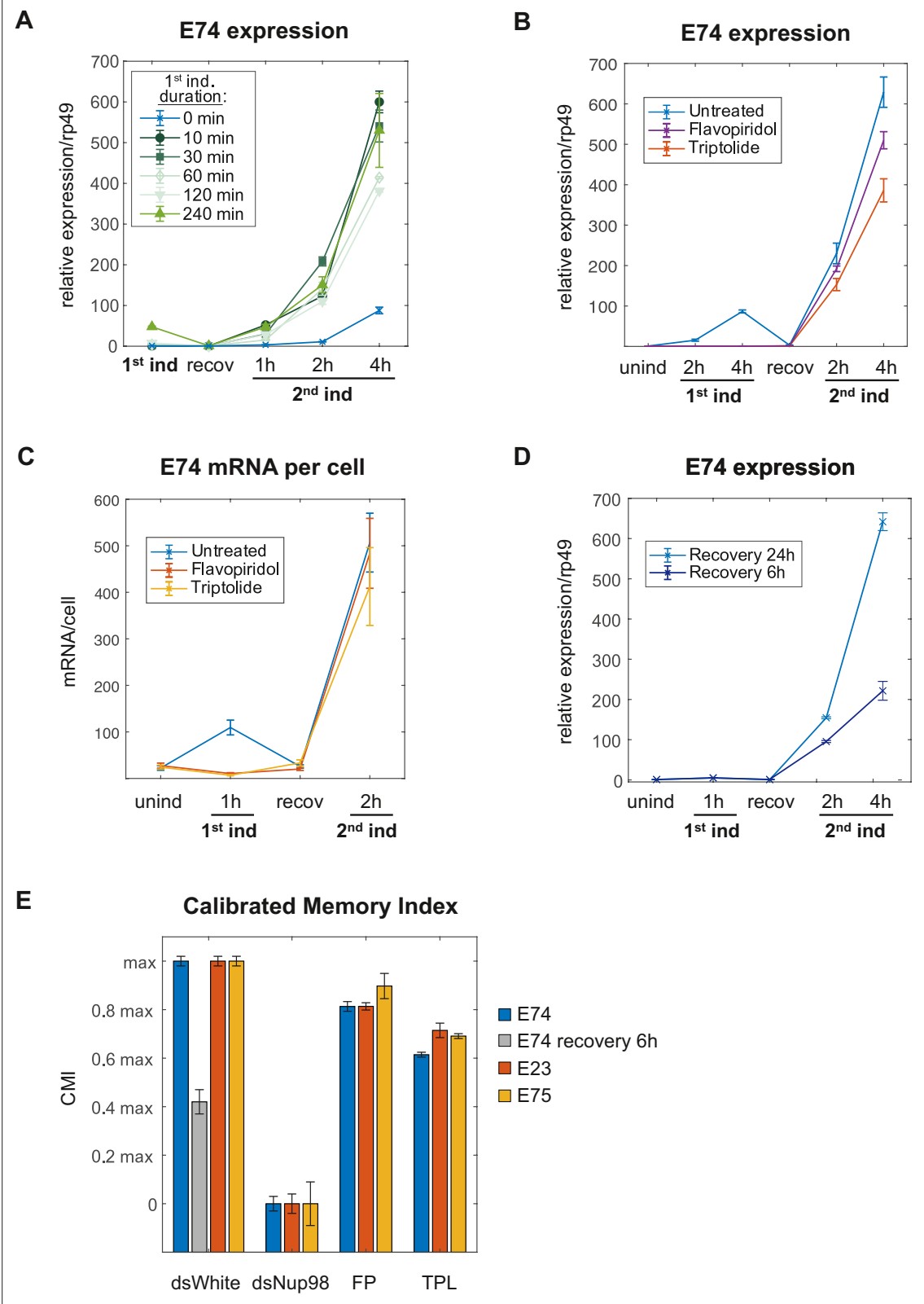

**Figure 5.** Induction of the memory state requires neither transcription nor prolonged 20E exposure. (**A**) *E74* mRNAs measured by qPCR and normalized relative to *rp49* with varying duration of 20E exposure during first induction. The data represent the mean of three independent experiments and the error bars the standard deviation of the mean. (**B**). Flavopiridol or Triptolide inhibitors were added 30 min prior 20E first induction at 1 μM and 5 μM, respectively. After 4 hr of induction, both, the hormone and the transcriptional inhibitors were washed-out and cells recovered for 24 hr. 20E re-induced

*Figure 5 continued on next page*

*Figure 5 continued*

cells were collected and, *E74* expression was measured by qPCR from three independent experiments. Fold change values were normalized using *rp49* and error bars represent the standard deviation of the mean. (**C**). Cells were treated with Flavopiridol or Triptolide as described in B and *E74* mRNAs were monitored by smFISH. Error bars represent standard deviation of the mean. (**D**). *E74* mRNAs levels were measured by qPCR and normalized against *rp49* using two different recovery times (6 hr and 24 hr). The data represent the mean of three independent experiments ± standard deviation. (**E**). Calibrated Memory Index (CMI) of *E74*, *E23*, and *E75* genes.

The online version of this article includes the following figure supplement(s) for figure 5:

**Figure supplement 1.** Establishment of the memory state is independent of transcription.

analyzed treatments demonstrates that while depletion of Nup98 has a substantial impact on transcriptional memory, inhibition of transcription by FP or TPL affects transcriptional memory to a much lesser extent, reducing the memory index on average by only 20–30% for any of the three genes (*Figure 5E*). Together, these results support our computationally derived model of transcriptional memory and reinforce the notion that ecdysone-driven induction initiates two independent events at a given locus: (1) the rapid switch and transition into active transcription, and (2) the slow Nup98-dependent conversion into the memory state (*Figure 6A–B*). The population dynamics of the transcriptional responses to ecdysone would thus reflect the changing mixture of the low-expressing and high-expressing/memory cells (*Figure 6A*), with Nup98 functioning as a key determinant for the prevailing fraction of high-expressing cells post-memory establishment. Our approach suggests that the main role of Nup98 in transcriptional regulation lies not with influencing transcriptional entry, but instead, with stabilizing and maintaining the specialized state of high transcriptional output, conversion to which is determined by a separate rate constant $k_C$ (*Figure 6B*).

## Discussion

The different models by which the primed state of transcription is established and maintained have been primarily drawn from bulk cell population studies. Using the *E74* gene as a model, we aimed to understand the gain of transcriptional priming in single cells within a population. Together, our data show that the acquired *E74* transcriptional memory is characterized by a high transcriptional output from a sub-population of cells transitioning into a memory state. Our modeling and experimental

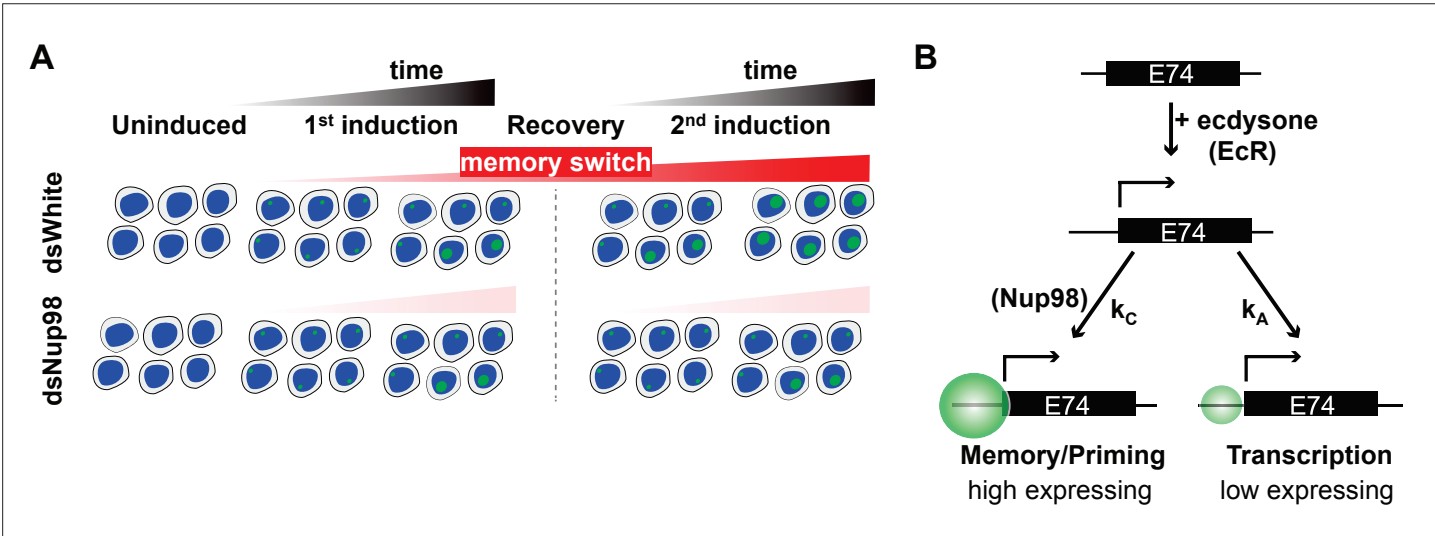

**Figure 6.** Memory switch model. (**A**) Two events are triggered independently by exposure to 20E: loci rapidly enter a low-expressing state and at the same time slowly switch from the default state to memory. Memory is characterized by high transcription rate in the presence of hormone, and importantly, cells continue to accumulate loci in the memory state after the hormone is withdrawn. When cells are exposed to 20E a second time, the converted memory loci transit into the high-expressing state, resulting in a more robust second induction. (**B**). Implication of the memory switch model: upon hormone exposure, loci engage in two separate activities, controlled by independent rate constants – entry into the active state (controlled by $k_A$) and transition into the memory state (controlled by $k_C$). In the memory switch model, normal levels of Nup98 are required for loci to accumulate and remain in the memory state ($k_C \gg k_{-C}$), such that depletion of Nup98 abrogates the maintenance of the memory state upon removal of hormone.

approaches suggest the following model (*Figure 6A–B*): upon ecdysone stimulation, two independent but simultaneous processes are initiated: (1) cells rapidly activate the ecdysone-dependent transcriptional response, characterized during this initial induction by a low rate of transcriptional output, and (2) concurrently, cells slowly progress into a specialized memory state defined by a high transcriptional rate, which is detected in subsequent ecdysone inductions. Importantly, we found that transition into the memory state is independent of the transcriptional process stemming from ecdysone activation since blocking transcription does not compromise the memory response significantly. Our transcriptional inhibition experiments suggest that memory establishment likely relies on ecdysone-triggered events that precede TPL-targeted TFIIH activity, such as pre-initiation complex (PIC) assembly, which was shown to remain intact upon TPL treatment (*Krebs et al., 2017*), or chromatin and architectural changes that accompany it. Another interesting conclusion from our model is that, if such a separation of transcription and memory occurs generally at inducible genes, then phenotypes of certain epigenetic regulators may not show up until much later on and would not be detected within a short time window, which is often used in Auxin-induced degradation system of rapid protein removal (*Morawska and Ulrich, 2013*; *Nabet et al., 2018*; *Nishimura et al., 2009*). As our findings reveal, on-going transcription and transcriptional memory can be controlled by separate mechanisms and would manifest their effects at different time scales.

A complex interplay of transcriptional factors, changes in histone modifications and the incorporation of histone variants have been linked to the transcriptional memory response of multiple genes. Additionally, transcriptional memory in certain systems involves the cytoplasmic inheritance of a regulatory protein produced during the initial process of transcription, such that the transcriptional memory response is dependent on the protein's level, which gets diluted in each cell division. This type of memory process has been found in yeast genes that respond to oxidative stress or changes in nutrients (*Guan et al., 2012*; *Kundu and Peterson, 2010*; *Zacharioudakis et al., 2007*). For example, in the case of the yeast *GAL1* gene, a key factor controlling the memory response is the passive inheritance of the trans-acting Gal1 protein itself, produced during transcriptional activation (*Kundu and Peterson, 2010*; *Zacharioudakis et al., 2007*). For the ecdysone-induced *E74* gene, this type of regulation seems unlikely since blocking all transcription still results in a robust memory response, demonstrating that the model of cytoplasmic inheritance described above does not extend to all memory systems.

Nup98 has emerged as an evolutionarily conserved factor required for transcriptional memory (*Light et al., 2013*; *Pascual-Garcia et al., 2017*). Our smFISH analysis, combined with mathematical modeling, revealed that active loci in a memory state initiate new transcription about 15 times faster than active non-memory loci. It also shed light on Nup98's role in this process, demonstrating that Nup98 promotes the rate of conversion between non-memory to memory state, once the activating agent has been removed. Thus, Nup98 does not affect transcription during initial induction, yet cells are unable to remain in or transition into the memory state upon hormone withdrawal, causing a defect in the amplified transcriptional response upon secondary ecdysone stimulation. How does Nup98 promote the memory state in molecular terms? In line with Nup98's role in securing the memory state, we have previously reported that Nup98 depletion disrupts the enhancer-promoter loops induced by ecdysone's initial induction (*Pascual-Garcia et al., 2017*). Our current results support a model where the stabilization of enhancer-promoter contacts by Nup98 might form part of the memory state, reinforcing our previously proposed notion that changes in enhancer-promoter contacts are functionally separate from initial transcriptional activity. Consistently, we have also reported that Nup98 gains physical interactions with EcR and architectural proteins upon ecdysone stimulation, suggesting that they might also form part of the memory state (*Pascual-Garcia et al., 2017*). In agreement with the present data, the memory complex formed by interactions of Nup98, architectural proteins, and enhancer-promoter contacts were similarly found to persist during transcriptional shut-off. Overall, our data are consistent with the idea that such architectural role of Nup98, as well as architectural functions suggested for other Nups (*Ibarra and Hetzer, 2015*; *Kuhn and Capelson, 2018*; *Kuhn and Capelson, 2019*; *Pascual-Garcia and Capelson, 2019*; *Sun et al., 2019*), may play a role in memory maintenance. It is worth noting that at the galactose-induce yeast gene *HXK1*, where memory is associated with gene loop interactions between the promoter and 3'-end of the gene, the maintenance of these loops is mediated by an NPC component Mlp1 and is thought to promote faster recruitment of RNA Pol II due to retention of transcription factors in the loop scaffold (*Tan-Wong et al., 2009*).

A similar mechanism may be executed by Nup98 in transcriptional memory of ecdysone-inducible genes, where a large complex, consisting of the looped gene, architectural proteins and key transcription factors, is 'locked in' by Nup98 to create a high-expressing memory state. In this manner, it is tempting to speculate that phase-separating properties of Nup98 and other Nups may be involved in creating a complex specialized for high transcriptional outputs (*Pascual-Garcia and Capelson, 2019*; *Schmidt and Görlich, 2016*; *Schmidt and Görlich, 2015*).

Another critical pathway implicated in transcriptional memory and in epigenetic memory in general is the deposition of histone modifications. H3K4 methylation is functionally linked to the transcriptional priming of a variety of genes (*Ding et al., 2012*; *D'Urso et al., 2016*; *Jaskiewicz et al., 2011*; *Light et al., 2013*; *Light et al., 2010*; *Muramoto et al., 2010*; *Sood et al., 2017*), and more recently, histone H3K36me3 was correlated with the acquisition of memory upon IFNβ stimulation in mouse fibroblasts (*Kamada et al., 2018*). Although the interplay between transcriptional memory and deposition of histone modifications has not been fully explored in the *Drosophila* system, multiple connections between H3K4 histone methyltransferases (HMTs) and factors that regulate ecdysone-induced transcriptional kinetics have been reported. For example, HMT Trithorax-related (Trr), responsible for the deposition of H3K4Me1, has been found to regulate the transcriptional activation of ecdysone-inducible genes through interactions with EcR (*Herz et al., 2012*; *Sedkov et al., 2003*). Additionally, Nup98 has been found to interact with the related HMT Trithorax (Trx) physically and genetically (*Pascual-Garcia et al., 2014*; *Xu et al., 2016*), and to regulate some of Trx target genes during fly development (*Pascual-Garcia et al., 2014*). Trx is a well-known regulator of epigenetic memory during development, and is critical for maintaining the active transcriptional state of homeotic genes, which define tissue identity (*Kingston and Tamkun, 2014*). Whether the discovered role of Nup98 in maintaining a specialized active state plays a role in the epigenetic memory of Trx targets remains to be determined, but our findings open an intriguing possibility that the transcriptional memory dynamics described here apply to the broader Trx-mediated memory.

Our modeling study indicates that the transition between non-memory to memory state is relatively long, lasting around 20 hr to convert 62% of loci (or 48 hr to convert 90% of loci) in an asynchronous cell population. In agreement with the model, our expression data show that reducing the time interval between ecdysone stimulations deteriorates the memory response. One possible explanation for such long-time scales of the $k_C$ rate constant is that the formation of a memory complex requires stochastic or biochemical events that involve these time scales. Another explanation may be a communication between cell-cycle completion and *E74* memory. The transmission of transcriptional states from mother to daughter cells has been studied in multiple organisms from human cells to developing fly embryos and is thought to be critical for the maintenance of differentiation programs (*Ferraro et al., 2016*; *Zhao et al., 2011*). In *Drosophila* embryos, the use of live imaging to visualize transcription uncovered a fourfold higher probability for rapid re-activation after mitosis when the mother cell experienced transcription (*Ferraro et al., 2016*). In mammalian cell culture, BRD4, a member of the Trx group (TrxG) proteins, has been implicated in faster re-activation kinetics of a previously activated transgene in post-mitotic cells (*Zhao et al., 2011*). In both cases, the authors proposed that transcription of the DNA template might render it more susceptible to rapid reactivation after mitosis, which may be achieved through heritable changes of DNA-bound transcriptional factors or histone modifications (*Ferraro et al., 2016*; *Zhao et al., 2011*). Additionally, it has been suggested that the process of mitosis itself is needed to help re-activate transcription to levels higher than those observed in the previous cell cycle, possibly via enhanced recruitment of regulatory factors to post-mitotic decondensing chromatin (*Zhao et al., 2011*). In this manner, it is plausible that the memory complex coordinated by Nup98 interplays with events immediately following mitosis to accelerate the dynamics of RNA synthesis, which could also explain why the enhanced transcriptional response depends on the length of time between ecdysone inductions.

## Materials and methods

**Key resources table**

| Reagent type (species) or resource | Designation | Source or reference | Identifiers | Additional information |
|---|---|---|---|---|
| Gene (*Drosophila melanogaster*) | Early gene 23 (*E23*) | Flybase | FBgn0020445 | |

*Continued on next page*

| Reagent type (species) or resource | Designation | Source or reference | Identifiers | Additional information |
|---|---|---|---|---|
| Gene (*Drosophila melanogaster*) | Ecdysone-induced protein 74EF (*E74*) | Flybase | FBgn0000567 | |
| Gene (*Drosophila melanogaster*) | Ecdysone-induced protein 75B (E75) | Flybase | FBgn0000568 | |
| Cell line (*D. melanogaster*) | S2-DRSC | *Drosophila* Genomics Resource Center (DGRC) | RRID: CVCL_Z992 | |
| Chemical compound, drug | 20-hydroxyecdysone (20E) | Sigma-Aldrich | Cat#: H5142 | |
| Chemical compound | Flavopiridol hydrochloride | Tocris Bioscience | Cat#: 3,094 | |
| Chemical compound | Triptolide | Sigma-Aldrich | Cat#: T3652 | |
| Chemical compound | Fugene HD | Promega | Cat#: E2311 | |
| Chemical compound | Trizol | Ambion | Cat#: 15596018 | |
| Commercial kit | Megascript T7 kit | Ambion | Cat#: AM1334 | |
| Commercial kit | Purelink RNA mini kit columns (; 12183018 A) | Ambion | Cat#: 12183018 A | |
| Commercial kit | QuantiTect RT-PCR | Qiagen | Cat#: 205,311 | |
| Commercial kit | PowerSYBR Green PCR Master Mix | Applied Biosystems | Cat#: 4367659 | |
| Chemical compound | Atto-565 | Sigma-Aldrich | Cat#: 72,464 | |
| Chemical compound | ProLong Gold mountant media | ThermoFisher | Cat#: P36930 | |
| MATLAB | | MathWorks | https://www.mathworks.com/ | |

## Cell culture and chemicals

We obtained *Drosophila* embryonic S2-DRSC cells from *Drosophila* Genomics Resource Center (DGRC) - this cell line is authenticated by DGRC and is one of the primary lines used by ModEncode and RNAi screens at DRSC (https://dgrc.bio.indiana.edu/product/View?product=181). We have confirmed mycoplasma negative status by PCR testing. Cells were grown at 25 °C in Schneider's medium (Gibco; 21720), supplemented with 10% (v/v) heat inactivated fetal bovine serum (Gibco; 10438034) and 1% (v/v) of penicillin-streptomycin antibiotics (10,000 U/mL) (Gibco; 15140163). For ecdysone induction experiments, 20-hydroxyecdysone (Sigma-Aldrich; H5142) was dissolved in 100% ethanol and used at 5 µM. Flavopiridol hydrochloride (Tocris Bioscience; 30-941-0) and Triptolide (Sigma-Aldrich; T3652) were prepared in 100% DMSO and added at 1 µM and 5 µM, respectively.

## dsRNA synthesis and transfection conditions

Double-stranded RNAs (dsRNA) fragments against *Nup98* or *white* genes were synthesized with Megascript T7 kit (Ambion; AM1334) using a PCR-ed DNA template from fly genomic DNA (primers listed in *Supplementary file 1*). The integrity of the RNA was assessed by running the denatured product in a 1.2% agarose gel. To knock-down cells we mixed 10 µg of dsRNA per million cells with 7.5 µl of Fugene HD (Promega; E2311) in serum-free media and we incubated the dsRNA cocktail with exponentially growing cells for at least 3 days.

## RNA extraction and quantitative PCR

Total RNA was isolated in 1 ml Trizol (Ambion; 15596018) and purified using Purelink RNA mini kit columns (Ambion; 12183018 A) following manufactured instructions. RNA concentration was determined by measuring absorbance at 260 nm with a NanoDrop 2000 (ThermoFisher; ND-2000). cDNAs were synthesized using one-step RT-PCR kit (Qiagen; 205311). To amplify specific cDNAs, primers were designed to span an exon-exon junction except for nascent *rp49* qPCR experiment, where primers were designed to amplify the first intron-exon boundary (primers listed in *Supplementary file 1*). In this case, the RNA was pre-treated with 1 U of DNAseI (ThermoFisher; EN0525) and incubated at 37 C for 30 min before reverse transcription reaction. We verified the efficiency of this digestion

by running control qPCRs with the DNAseI-treated RNA samples, and checked for the presence of insufficiently digested genomic DNA.

For absolute quantification analysis, the template for *E74* synthetic RNA standard was generated from a cDNA library and amplifying with specific primers covering the first two exons of *Eip74EF-RA*. ssRNA was in vitro transcribed using the Megascript T7 kit. The concentration in grams per volume of the synthesized reference RNA was determined by fluorometric quantification using the RNA HS assay kit (ThermoFisher; Q32852) and Qubit 2.0 (ThermoFisher; Q32866). RNA dilution series were converted to copy numbers per volume using the molecular weight of the RNA standard (MW = 126060 g/mole). For the RT reactions of the RNA standard dilutions, we added an additional *Eip74EF*-RA primer in addition to the random primers included in the reaction kit. To minimize our experimental error, we also constructed a DNA standard using *E74* specific primers. We used this standard to calculate the amplification efficiency of the *E74* primers by fitting a linear curve to $\log_2$(Ct) as a function of *E74* DNA template copy number; the resulting line contains a slope of –3.566 and $R^2$ of 0.9985. We constructed the RNA standard curve by single-parameter fit of a line of –3.566 slope to the previously retrotranscribed RNA dilution series. We also factored the losses of mRNA extraction and purification by adding a known amount of in vitro transcribed *E74* RNA to Trizol followed by purification and comparing the obtained amount of RNA; this approach resulted in 56% of RNA losses.

In all qPCR experiments, we used PowerSYBR Green PCR Master Mix (Applied Biosystems; 4367659) and QuantStudio 7 Flex thermal cycler (ThermoFisher; 4485701). Each qRT-PCR was repeated at least three times, the values were normalized to the *rp49* transcript unless otherwise stated, and the error bars represent the standard deviation of the mean.

To compare the impact of inhibiting transcription during the 20E reinduction between different ecdysone-responsive genes, we define the calibrated memory index (CMI) as the ratio of the Nup98-dependent memory in inhibitor-treated cells to untreated cells:

$$CMI = \frac{F * M_I}{R}$$
$$F = 1 - M_N / M_C$$
$$R = M_C - M_N,$$

where $M$ is the fit slope of the second induction response either in Nup98-depleted cells ($M_N$), in control cells ($M_C$), or in the presence of transcriptional inhibitors ($M_I$). $F$ represents the Nup-dependent fraction of the second induction response, whereas $R$ is the absolute value of the Nup98-dependent fit slope upon second induction and is the value to which $F * M_I$ must be normalized for comparison among genes. The maximum and minimum of the responses are specific to each gene because each gene exhibits a different value for the increased slope in untreated cells and the extent of reliance on Nup98. For convenience, CMI is zero under Nup98-depletion and the maximum is 1. CMI therefore represents the fraction of memory that remains during the second induction upon transcriptional inhibition during the first induction.

## Single-molecule RNA fluorescence in situ hybridization (SmRNA FISH)

S2 cells were cytospun in poly-L-lysine treated coverslips and fixed with 4% para-formaldehyde for 10 min. Coverslips were rinsed 3 X in PBS and submerge in cold 70% EtOH for at least 24 hr. Complementary probes to the reading frame of *eip74ef*-RA were designed using Stellaris Probe Designer (https://www.biosearchtech.com/stellaris-designer), ordered from Biosearch and conjugated to Atto-565 (Sigma-Aldrich; 72464). Cells were washed twice with wash buffer [2 X SSC, 10% formamide, 0.01% Tween-20] and equilibrated with hybridization buffer [2 X SSC, 10% formamide, 10% dextran sulfate,1 µg/ml BSA]. The hybridization to probes was performed overnight at 37°C in a moisturized chamber at a concentration of about 1 nM. After hybridization, samples were washed 3 X in prewarmed wash buffer and incubated at 37°C for 30 min. We performed a final wash with 2 X SSC, stained with Hoechst and mounted in ProLong gold (ThermoFisher; P36930). Imaging was performed by laser-scanning confocal microscopy on a Leica SP8 with a 63 x oil immersion objective using identical scanning parameters and laser power for all samples. Voxel dimensions are 76 × 76 × 250 nm.

## Image Analysis

### mRNA detection and normalization

By scanning confocal imaging, RNA puncta corresponding to single mRNAs, RNPs, and sites of nascent transcription all appear as diffraction-limited objects whose fluorescence intensities correlate with mRNA content (*Little et al., 2015*; *Little et al., 2013*; *Little et al., 2011*). Puncta were separated from imaging noise and centroids of true diffraction-limited objects were found with difference-of-Gaussian (DoG) thresholding using custom MATLAB scripts (*Little et al., 2013*) on deconvolved images followed by fluorescence intensity measurements using raw images as described (*Little et al., 2015*). Objects were classified as mRNA puncta or nascent transcription sites on the basis of DoG intensities as described (*Zoller et al., 2018*). Both the numbers and fluorescence intensities of non-nascent puncta increase during hormone induction. To obtain measurements of mRNA content of each individual puncta in absolute units by smFISH, we adapted a prior normalization technique (*Little et al., 2015*; *Little et al., 2013*; *Little et al., 2011*; *Zoller et al., 2018*). Briefly, we divide the fluorescence intensity of all puncta by the mean intensity of the non-nascent site puncta found during uninduced conditions, since, in the absence of hormone, the mean number of mRNAs detected by smFISH corresponds to the mean number measured by qPCR. Individual puncta are thus assigned a value corresponding to the equivalent number of finished, mature mRNAs they each contain. Because non-nascent puncta are mostly found in the cytoplasm, we term the units of this measurement 'cytoplasmic units' or C.U.s, the unit intensity of single mRNAs (*Little et al., 2013*). These measurements are thus in absolute units. When reporting transcriptional activity, we sum the fluorescence of all nascent site puncta assigned to each cell (*Little et al., 2013*; *Zoller et al., 2018*). This is advantageous because of the phenomenon of chromosome pairing, prevalent in *Drosophila*, in which homologous chromosomes are found in close physical association (*Joyce et al., 2016*). Pairing prevents unambiguous assignment of fluorescence to individual nascent sites, but does not preclude us from accurately assessing transcriptional activity in single cells. Objects were designated as sites of nascent transcription using a threshold of three times the intensity of single mRNAs. For nuclei that contain fewer than three objects that are greater than 3 x the average intensity of cytoplasmic spots, 1 or two spots are selected at random and assigned the status of transcription sites; the random assignment yields a more accurate assessment of transcription than an assertion that no transcription occurs, as previously described (*Zoller et al., 2018*). All transcriptional activity reported in *Figures 3 and 4* is in units of C.U. per individual cell. In contrast, all parameter values derived from fitting are for individual loci, as described below.

### Nuclear-cytoplasm segmentation

Hoechst stain was used to determine pixels corresponding to nuclear volumes as described (*Petrovic et al., 2019*), and the same approach was applied to the low-level nonspecific cytoplasmic fluorescence in the RNA channel to delineate total cell volumes. All RNA puncta were assigned to the nearest nucleus using on the basis of nearest-neighbor comparisons of the positions in 3D space of the centroids of puncta and nuclei, as described (*Petrovic et al., 2019*).

## Modeling

### mRNA degradation

mRNA stability was assessed for the first and second inductions by collecting cells for qPCR at 30 min intervals. After 4 hr of hormone treatment, cells were washed-out and treated with 1 µM Flavopiridol to disrupt transcription. qPCR was used to measure *E74* levels relative to *rp49* as a function of time following the start of transcription inhibition. qPCR was performed in triplicate on cells treated for 72 hr with dsRNA against *Nup98* or *white* genes. The data were fit to a model of exponential decay using nonlinear regression to obtain mRNA lifetimes and 95% confidence reported in *Figure 2B* and *Figure 2—figure supplement 1B*.

### mRNA export

Image segmentation of cells into nuclear and total cell volume described above was used to assign all non-nascent RNA puncta to either the nucleus or cytoplasm based on the centroid positions of puncta in three dimensions. Puncta densities in both volumes were calculated as the number of puncta per cubic micron, and the fraction of puncta found in cytoplasm calculated for individual cells. Under

an assumption of unchanging mRNA degradation, the ratio of cytoplasmic to nuclear densities is constant regardless of expression level as long as RNA processing and transport are rapid compared to mRNA degradation (*Aleman et al., 2021*). Since degradation rates are slow and indistinguishable between Nup98 knockdown and control, we conclude that mRNA transport rates are unaffected by Nup98 depletion.

## mRNA accumulation and production

Fitting of qPCR data was performed using nonlinear regression and the measured mRNA degradation rate to obtain 95% confidence intervals for parameter values. Fits were performed for each experimental condition individually to produce the values displayed in the panels. S2 cells are tetraploid with a division time of 24 hr, spending similar amounts of time in G1 and G2 (*Cherbas and Gong, 2014*). We therefore performed all qPCR fits using the assumption that the average number of *E74* loci per cell is six. All fit parameter values are reported in terms of individual loci. Curves shown in *Figure 1B* are piecewise polynomials. Models in *Figure 2* each contain a single free parameter representing either the rate of constant mRNA production or the rate of linear increase in the mRNA production rate as a function of time. Both models in *Figure 3* contain two free parameters. For the model of constant RNA Pol II loading, these are the first-order rate of conversion to the active state $k_A$ and the rate of RNA Pol II loading while active $k_{Pol}$. For the model of accelerating $k_{Pol}$, $k_A$ is fixed so as to virtually guarantee that all loci convert to the active state in 1 min (as noted in the text, *Figure 3— figure supplement 1D* is plotted with a much smaller $k_A$ for illustrative purposes). We introduced an additional parameter, the RNA Pol II footprint on the DNA template, to provide a natural limit on the maximum attainable $k_{Pol}$.

To compare the fitting of the qPCR results to the measurements of transcriptional activity obtained by smFISH, the rates from fitting were utilized in Monte Carlo simulations of transcription (*Gillespie, 1976*). We simulated the time-dependent evolution of RNA Pol II numbers and positions on the *E74* gene for 100,000 cells for each of the four conditions (Nup98 knockdown or dsWhite/control, with or without 20E). The time scale of the simulation was set by the previously measured RNA Pol II elongation rate in S2 cells of 1500 nt/min (*Izban and Luse, 1992*; *Ardehali and Lis, 2009*; *Buckley et al., 2014*; *Yao et al., 2007*). The simulation was updated every 1/1500 min, the interval needed to transcribe one nucleotide. For the model of increasing production rate, the production rate was also updated based on the acceleration parameter multiplied by the elapsed time. RNA Pol II molecules and newly completed mRNA molecules were assumed to be evicted as soon as transcription was finished. For simplicity, we assumed that half of simulated cells are in G1 and half in G2, therefore containing either four or 8 *E74* loci. We therefore simulated 600,000 single loci and combined them randomly into 50,000 sets of 4 and 50,000 sets of 8 to represent 100,000 individual cells. We converted RNA Pol II positions into C.U.s by noting that the RNA Pol II position in the gene body determined how many probe binding sites are present in the nascent RNA. The total C.U.s per cell was attained by summing the number of probe binding sites present across all four or eight simulated loci. Because all probes bind exonic sequences, neither the simulations nor observations are affected by splicing. With this procedure, we deduced parameter values in terms of single loci, rather than single cells. Starting conditions assigned to each cell were chosen at random from a Poisson distribution using the mean number of detected puncta under uninduced conditions. Simulated transcriptional activity values were taken at appropriate times after the start of the simulation (60, 120, and 240 min) to compare the simulated distribution to that observed by smFISH. Goodness-of-fit scores were calculated by finding the intersection of the areas under the normalized histograms of simulated and observed cells, using histograms generated with bins of identical width. For convenience, we compared log(C.U.) values due to the long tail of observed transcriptional activity, identical to a previously utilized approach for describing the distribution of mRNAs in ribonuclear protein complexes (*Little et al., 2015*). The areas of intersection between simulated and observed histograms for each of the three time points corresponding to each of four conditions were summed and then divided by the area of the union, generating an effective Jaccard index (JI) that was used as a goodness-of-fit score. A score of 1 indicates perfect overlap between observation and simulation. We note that the absolute value of the goodness-of-fit score for any individual model is not by itself informative and is heavily reliant on the positions of bin edges; in contrast, the comparison of scores between identically prepared distributions informs on the extent of difference in overlap between model and observation.

For the four-parameter model of transcriptional memory (*Figure 4*), we searched parameter space across eight orders of magnitude: For the first-order activation and conversion rates $k_A$ and $k_C$, between $10^{-8}$ and 1 min$^{-1}$, and for the $k_{Pol}$ rates, between $10^{-7}$ and 10 min$^{-1}$. Fitting the curves derived from qPCR was uninformative, as a vast volume of parameter space can describe population-averaged data. We therefore performed Monte Carlo simulations of transcription in order to approximate the observed distributions of transcriptional activity at all six time points in control conditions. In principle, this entails a search through 4D space across cells simulated for 32 hr (4 hr 1st induction, 24 hr recovery, and 4 hr 2nd induction). However, the combination of the four parameter values is constrained to match the trend of the average mRNA production rate determined by the prior fit of the qPCR results. This effectively reduced the search space to three dimensions, since we could choose a value for one parameter and simulate across combinations of the remaining three. To reduce computational burden, we simulated 10,000 cells for each parameter set over the four hours of the first induction. We then inferred the fraction of loci that would be in the memory state with a given $k_C$ following a 24 hr recovery period, and randomly assigned memory status to that fraction of loci, in order to avoid simulating the recovery period. We then simulated the 4 hr period of the second induction. The histograms of simulated and observed C.U.s were then scored by JI. This rapidly narrowed the possible range of parameter values to within approximately one order of magnitude. Subsequent fine-grained search of parameter space employed 100,000 cells per parameter set to find the maximum JI. The 95% confidence interval was assigned by bootstrap, taking 1000 random subsets of single cells from the data, redeploying the search of the narrow parameter window, and finding the mean and two standard deviations of the parameter values producing the maximum JI.

We note that this procedure requires that the transcriptionally active state has a lifetime significantly shorter than 24 hr after hormone withdrawal. We did not attempt to measure the lifetime of the active state. However, we note there is minimal transcription following 24 hr recovery, supporting an assumption of short active lifetime without hormone. Likewise, we did not attempt to measure the lifetime of the converted state, which would only become apparent in Nup98 knockdown cells upon hormone withdrawal. However, the fit parameters from control data provide a reasonable explanation of both control and Nup98 knockdown cells. This supports the hypothesis that the memory state lifetime is significantly less than 24 hrs in the absence of Nup98, whereas memory is maintained indefinitely in normal cells and their progeny.

## Acknowledgements

We are thankful to members of the Capelson, Little, DiNardo and Bashaw laboratories for insightful comments and suggestions. We are also grateful to Jack M Gallup (Newport Laboratories) for helping with qPCR analysis, Jessica Talamas (Dana Farber Cancer Institute) and Terra Kuhn (European Molecular Biology Laboratory) for critical reading of the manuscript. We acknowledge the *Drosophila* Genomics Resource Center, supported by National Institutes of Health grant 2P40OD010949, and the Cell and Developmental Biology Microscopy Core Facility for confocal microscope use. M C is supported by the National Institutes of Health grant R01GM124143.

## Additional information

### Funding

| Funder | Grant reference number | Author |
| --- | --- | --- |
| National Institutes of Health | R01GM124143 | Maya Capelson |

The funders had no role in study design, data collection and interpretation, or the decision to submit the work for publication.

### Author contributions

Pau Pascual-Garcia, Conceptualization, Formal analysis, Investigation, Methodology, Validation, Visualization, Writing – original draft, Writing – review and editing; Shawn C Little, Conceptualization,

Data curation, Formal analysis, Investigation, Methodology, Software, Writing – original draft, Writing – review and editing; Maya Capelson, Conceptualization, Formal analysis, Funding acquisition, Investigation, Supervision, Writing – original draft, Writing – review and editing

**Author ORCIDs**
Maya Capelson (ID) http://orcid.org/0000-0002-1695-316X

**Decision letter and Author response**
Decision letter https://doi.org/10.7554/eLife.63404.sa1
Author response https://doi.org/10.7554/eLife.63404.sa2

## Additional files

### Supplementary files
• Supplementary file 1. List of primers used in this study.

• Transparent reporting form

### Data availability
No sequencing or structural datasets have been generated by this study. All of the data analyzed during this study is included in the manuscript.

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
