## [Editor Report]

The authors quantitatively assess transcriptional memory in the context of mathematical modeling and testing of the models through single cell approaches. They extend their work to show how single cell data relates to population-level transcription outcomes. The models produced make predictions that the authors successfully test to demonstrate that transcription initiation is not necessary for establishment of memory.

---

## [Decision Letter]

**Decision letter after peer review:**

Thank you for submitting your article "Nup98-dependent transcriptional memory is established independently of transcription" for consideration by *eLife*. Your article has been reviewed by 4 peer reviewers, and the evaluation has been overseen by a Reviewing Editor and Kevin Struhl as the Senior Editor. The reviewers have opted to remain anonymous.

The reviewers have discussed the reviews with one another and the Reviewing Editor has drafted this decision to help you prepare a revised submission.

If it is possible to carry out the suggested experiment on transcriptional inhibition, we urge you to do so.

In this manuscript by Pascual-Garcia et al., the authors use single-cell gene expression analysis to dissect transcriptional memory in the ecdysone response. They explore several different models and the surrounding parameter space in order to fit the data and then arrive at a plausible multi-state model for how Nup98 might be enabling transcriptional re-activation. The overall conclusion is that ecdysone stimulation in the first round 'primes' the re-activation state through a Nup98-dependent process. This priming does not depend on transcriptional activation because treatment with the elongation inhibitor flavopiridol doesn't interfere with the memory response.

All the reviewers find the work interesting and potentially important. However before it can be published, they had several concerns in common. One of them deals with the clarity of the manuscript, in particular the explanation of the modeling and why certain states were chosen, and why so much text deals with the non-working model.. A second concern is that the quality of the FISH on which the approach is based is not optimal. A third concern, as expressed by all reviewers (esp #4) is that the use of the drug flavopiridol does not test the hypothesis that transcription is not necessary for memory because it acts primarily on elongation and not initiation, where chromatin modifications are made. Further data is requested using a drug that acts earlier in the transcription. Finally reviewer #1 questions the conclusion that RNA turnover is not a factor given the time for memory to occur.

The full reviews are below. We feel that the essential experiment would be to determine which component of transcriptional inhibition (initiation, escape) is essential or not for memory.

*Reviewer #1:*

This is a combined modelling and experimental study probing the nature of the ecdysone induced memory response in S2 cells. This is a very interesting biological problem, and detailed mechanisms- beyond data indicating that protein X or modification Y are required, have proved elusive outside the yeast GAL system. As such, studies probing the causality underlying transcription memory phenomena are welcome.

In the study, the simple models of transcriptional dynamics are discarded, with reference to qPCR and smFISH data and measurements of export and RNA turnover. The researchers fix on a 4 state model that fits the data well, then test 2 predictions of the model- based on the requirement for transcription and the timescale of the memory phenomenon.

The work appears well performed, and for the most part clearly documented, and should certainly be part of the scientific record in close to its current form.

My major reservation is the state of the field. This paper seems largely a model optimisation paper, and the work appears very well carried out. However, there are many different multi-state models around these days, all specific to different genes and systems. From the perspective of understanding transcription dynamics- does this carry us further? From the perspective of memory, it is interesting the transcriptional elongation at the locus is not initially required- but the state change of the cell, or gene, that brings about the memory is no clearer beyond this.

One issue that needs to be addressed before publication relates to the RNA turnover aspect. In the experiment in Figure 2B this analysis is conducted at 4h not 24h (when the initial memory effect was detailed in Figure 1), so is your conclusion about no role for RNA turnover relevant to the memory phenomenon? Especially in the light of Figure 7, where both the experiment and model indicate a longer timescale is required to get the memory effect.

*Reviewer #2:*

Transcriptional memory is a phenomenon whereby certain inducible genes show stronger/faster induction if they have previously been expressed. This behavior can persist through multiple cell divisions. Such phenomena have been explored in yeast, plants, flies and mammalian cells and several different types of memory have been discovered. In the cases where we have some mechanistic understanding, there are two classes of memory. The regulation of some genes is impacted by the persistence of transcriptional regulators that were previously produced and cytoplasmically inherited (e.g. yeast GAL1). The second class of genes (e.g. yeast INO1 and interferon γ-induced genes in HeLa cells) there are cis-acting changes in transcription factor binding and/or chromatin structure that impact future expression. This latter class of genes shows a physical interaction with the nuclear pore protein Nup98 and memory requires Nup98.

This manuscript explores the transcriptional memory of ecdysone-regulated genes in *Drosophila* S2 cells, a phenomenon that is Nup98-dependent and correlates with a Nup98-modulated enhancer-promoter loop. The focus of the study is a thorough and quantitative anlalysis of the population dynamics of transcription during the initial induction and memory phases using single molecule RNA FISH. These data are then used to assess three different quantitative models for memory: 1. A two-state model in which memory increases the number of cells that respond to ecdysone, 2. A two-state model in which memory increases the rate of RNA polymerase loading onto the promoter and 3. A four-state model in which these genes exist in an equilibrium between two different expression states (low and high) and this equilibrium is regulated by memory. The authors find that model 3 is the best fit for the observations and, importantly, this model makes two predictions that they could test. First, this model predicts that transcription should not be required for memory. Second, memory should "develop" following the initial induction over some time scale. The authors then test both of these predictions and find that they are true. The former is interesting and rules out that the production of a transcriptional regulator during the first induction is key to this process. The latter is very interesting and surprising and will stimulate additional work to discover its meaning. Overall, the paper is interesting and impactful and worthy of publication in *eLife*. I have some suggestions for clarifying changes to the manuscript and, if possible, additional experiments that would strengthen the story.

1. The four state model is not well described in the manuscript and the schematic in Figure 5A is confusing. This schematic suggests that there are equilibria between non-memory and memory (which is understandable) and between low-expressing and high-expressing states (also understandable) but that there is also an equilibrium between non-memory and low-expressing and between memory and high-expressing, which does not make sense to me. I think what the authors are trying to convey is that the equilibrium between low-expressing and high-expressing states is influenced by Nup98-dependent memory. If so, then the model should include the following states: non-memory, low-expressing, non-memory high-expressing, memory low-expressing and memory high-expressing. Between each of these states would be an equilibrium and the equilibrium constants for the equilibrium between low and high expressing states would be different for non-memory and memory. I assume that this is how they generated their model. If not, please spend more space to clarify the model and provide an explanation of how the non-memory state is in equilibrium with the low-expressing state, etc.

2. The technical perfection of smRNA FISH is unclear. I doubt it is perfect. Yet, the modeling is highly leveraged on the goal of matching the smRNA FISH data. Some additional controls would strengthen the interpretation. For example, the untreated S2 cells (0h) should be included for the smRNA FISH and modeling. Likewise, knowing how control genes that are not up/down-regulated by ecdysone behave in smRNA FISH would improve the rigor of this approach.

3. Some molecular information about transcription factor occupancy or chromatin changes at the enhancer that is involved in the memory-enhancing loop would greatly strengthen the mechanistic insights of this paper.

*Reviewer #3:*

In this manuscript by Pascual-Garcia et al., the authors use single-cell gene expression analysis to dissect transcriptional memory in the ecdysone response. They explore several different models and the surrounding parameter space in order to fit the data and then arrive at a plausible multi-state model for how Nup98 might be enabling transcriptional re-activation. The overall conclusion is that ecdysone stimulation in the first round 'primes' the re-activation state through a Nup98-dependent process. This priming does not depend on transcriptional activation because treatment with the elongation inhibitor flavopiridol doesn't interfere with the memory response. Although there is not much biochemical insight, I find this conclusion interesting. Overall, the analysis is fairly rigorous, and I applaud the effort to apply more quantitative models to the field of transcriptional memory.

1. The modeling is a mix of ordinary differential equation analysis and stochastic simulations with the Gillespie algorithm. It is a little unwieldy at times, and I feel the manuscript spends quite a bit of time on a model which doesn't work. Since this non-working model is the authors' invention (and not a standard in the field per se), it seems a bit awkward to present the story this way. Ideally, all the models would be stochastic models from the beginning since the single-cell analysis is the main thrust at the end. Although I find their approach a little ad hoc given the current state of the field, I don't feel strongly enough about it to demand a more rigorous treatment.

2. The persistence of memory in the presence of flavopiridol is a striking result. Would the authors predict a similar result for an inhibitor working upstream of transcriptional pause release? Specifically, the authors mention in the discussion that enhancer-promoter loops may be implicated. One experiment which the authors might consider (which I am not suggesting as necessary for a revision) is whether antagonists or partial agonists can uncouple these two activities of ecdysone. In other words, they have a negative result which emerges as a model prediction but not a positive result.

*Reviewer #4:*

The authors use a combination of RT-qPCR, smFISH and mathematical modeling to probe the mechanism and contribution of NUP98 to transcriptional memory upon hormone stimulation in *Drosophila* cells. The authors reveal that the transcriptional memory response is independent of transcription levels immediately after hormone treatment. Furthermore NUP98's role in transcriptional memory relates to its ability to stabilize a slow transition (~20 hrs) of the gene from low-expression to high-expression state. While the results are quite striking and the interpretations are timely in some regards, a few controls and clarifications in the text are necessary before recommending publication in *eLife*.

1. The authors rely heavily on smFISH to validate their computational modeling. However, the accuracy of quantitation of the smFISH is not evident. The authors should show a histogram of intensities of single diffraction limited spots along with a representative figure of multiple zoomed in panels of the single RNA spots. This is important as the majority of spots, particularly in the transcriptional memory response, are not diffraction limited which complicates both the presentation and analysis of their data.

2. Is there any evidence of E74 being alternatively spliced which may differentially affect FISH probe binding to the mRNAs producing large heterogeneously sized diffuse spots seen in the figure 1D?

3. While the differences are readily apparent in most of the data, the authors do not show any statistical testing in many if not most plots (e.g. violin plots Figure 1E). This should be corrected in a revised manuscript.

4. S2 cells are considered aneuploid. For nascent transcription site analysis, it would be helpful for the readers to see a simple diagram showing the percentage of cells showing 1, 2, 3, 4 etc active TS sites throughout the time course as it was not apparent whether a certain percentage of cells in the population never expressed E74.

5. Figure 7 is one of the most key figures that the authors use to conclude that NUP98 is stabilizing a memory state independent of transcription. Yet they have chosen to use a less informative assay (e.g. RT-qPCR) rather than the most sensitive assay (e.g smFISH) to support their model and conclusions. The authors would make a stronger case for their findings if they employed smFISH to monitor transcript abundance and TS activity after inhibiting transcription.

6. FP primarily only prevents Pol II pause escape and not promoter escape of Pol II. So transcription initiation and Pol II loading at the promoter still occurs in flavopiridol treated cells. This is important because chromatin marks associated with active transcription are likely still being placed in the promoter and enhancer regions of E74 even though a complete transcript is not being produced. The authors should repeat their assays and treat the cells with inhibitors such as triptolide or THZ1 that act earlier in the transcription cycle or better yet use an ecdysone receptor mutant that prevents Pol II recruitment or chromatin modification to the gene in the first place.

[Editors' note: further revisions were suggested prior to acceptance, as described below.]

Thank you for submitting your article "Nup98-dependent transcriptional memory is established independently of transcription" for consideration by *eLife*. Your article has been reviewed by 3 peer reviewers, and the evaluation has been overseen by a Reviewing Editor and Kevin Struhl as the Senior Editor. The following individuals involved in review of your submission have agreed to reveal their identity: Jonathan R Chubb (Reviewer #1); Jason H Brickner (Reviewer #2).

All reviewers find that the manuscript is novel and exciting. However reviewer #4 has additional comments that need to be addressed, in particular the observation 1 that triptolide is not required for PIC formation in *Drosophila*. This issue needs to be addressed in the Discussion along with other issues raised by this reviewer. Please send a response to these comments with the revised manuscript.

*Reviewer #1:*

This is a much improved manuscript, and deals extensively with the comments raised by myself and the other reviewers. The additional experiments, particularly the more comprehensive RNA turnover work, and the transcription initiation blockade, are a great addition. I think that the overall message is much clearer in the revised paper, with the condensed version of the modelling much more straightforward to read. The overall message, that the transcriptional process is not required for the sensitisation, is now much more clear than I found in the initial submission, and is an important addition to the cellular memory literature.

*Reviewer #2:*

The authors have responded to my suggestions and, in some cases, added additional data or clarification to the paper. The manuscript is now stronger and publishable in *eLife*.

*Reviewer #4:*

The authors take an important step in quantitatively assessing transcriptional memory in the context of mathematical modeling and testing of the models through single cell approaches. Furthermore, the authors extend their work to show how single cell data relates to population-level transcription outcomes. This second approach is highly relevant since transcriptional memory has been traditionally studied at the population-level allowing interpretation of previous studies in the context of single cell data. More importantly, the models produced can make predictions that the authors successfully test to show that transcription initiation is not necessary for establishment of memory. The authors do a good job of addressing the reviewer's critiques. While a clear mechanism for Nup98 establishment of memory is still lacking, I think that this manuscript represents an important step in integrating single cell data into studying memory. There are a few critiques that should still be addressed before final acceptance for publication.

1. While the authors indeed show that transcription initiation is not required for memory, Triptolide treatment does not prevent PIC formation in *Drosophila* (Krebs et al., 2017, Mol Cell, which the authors cite in the manuscript). PIC assembly actually increases during Triptolide treatment suggesting that key factors and histone marks related to the assembly of the PIC (e.g. H3K4 methylation due to its presumed role in recruiting TFIID) are likely still present and capable of establishing memory in the absence of initiation. The authors should be very clear of this possibility in the discussion.

2. Lines 175-177 the authors are essentially making a statement about transcriptional noise being independent of Nup98. It would be highly beneficial to quantitatively measure transcriptional noise (CV2 or std2/mean2) under the different conditions to validate this statement.

3. I realize that Models 1 and 2 are not viable in the end. However, the wording in Lines 301-306 is misleading creating a potential flaw in these models that seems to be carried throughout the manuscript. The authors invocation of a quantitative assessment of the PIC (e.g. Author's should use the term PIC since RNA Pol II holoenzyme can be confusing) footprint into the model is naive and underestimates the role of a stable TFIID scaffold and TFIIB in re-initiation rates (Zhang et al. Genes and Dev 2016, Yudkovsky et al., Nature 2000). It is highly likely that core promoter accessibility plays a major role in going from an inactive to active state. But core promoter accessibility would likely be a binary effect and not scaleable as the authors seem to intimate in lines 302-304 and let freely fluctuate in their modeling. Model 1 and 2 also assumes that transitions to the inactive states are negligible suggesting that the gene is stuck in the ON state and therefore Nup98 is enhancing Pol II's attempted loading onto a stable PIC scaffold of defined size (e.g. footprint).

4. It would be interesting to see whether the FP and TL treatments change the distributions of transcriptional output during the second induction in single cells compared to those shown in Figure 4.

---

## [Author Response]

The full reviews are below. We feel that the essential experiment would be to determine which component of transcriptional inhibition (initiation, escape) is essential or not for memory.

We thank the Editors for a positive assessment of our work, and for outlining the main suggestions by the reviewers and the key experiment needed for revisions. As we detail further below, we have addressed all 4 main points, either experimentally or via further explanation, and have added the essential experiment utilizing an additional transcriptional inhibitor that acts earlier in the transcriptional process.

Reviewer #1:This is a combined modelling and experimental study probing the nature of the ecdysone induced memory response in S2 cells. This is a very interesting biological problem, and detailed mechanisms- beyond data indicating that protein X or modification Y are required, have proved elusive outside the yeast GAL system. As such, studies probing the causality underlying transcription memory phenomena are welcome.In the study, the simple models of transcriptional dynamics are discarded, with reference to qPCR and smFISH data and measurements of export and RNA turnover. The researchers fix on a 4 state model that fits the data well, then test 2 predictions of the model- based on the requirement for transcription and the timescale of the memory phenomenon.The work appears well performed, and for the most part clearly documented, and should certainly be part of the scientific record in close to its current form.My major reservation is the state of the field. This paper seems largely a model optimisation paper, and the work appears very well carried out. However, there are many different multi-state models around these days, all specific to different genes and systems. From the perspective of understanding transcription dynamics- does this carry us further? From the perspective of memory, it is interesting the transcriptional elongation at the locus is not initially required- but the state change of the cell, or gene, that brings about the memory is no clearer beyond this.

We thank the reviewer for the overall positive assessment of our manuscript and for considering it a welcome addition to the field. While it is certainly the case that many models of transcription have been put forth, the memory field has suffered from a lack of quantitative approaches. We believe the simplicity of our model provides a general framework against which to test the behavior of genes in the diverse systems that exhibit the memory phenomenon. We also reemphasize our use of modeling to generate novel predictions, which we have tested, unlike many papers focused only on model optimization.

Although we agree that our findings do not pinpoint specific state changes or molecular factors that define transcriptional memory, we believe they do offer important conceptual insight into how memory works and into how we interpret effects of transcriptional regulators. Our key conclusion that transcriptional elongation and memory establishment appear to be separate pathways will direct and restrict future inquiries into possible players of transcriptional memory. For instance, our studies allow to discount a number of previously suggested players such as transcription-coupled histone modifications or synthesized mRNA and guide the field to focus on a much more targeted set of events surrounding or preceding transcriptional initiation. Similarly, our model predicts that loss of function phenotypes of many epigenetic regulators may not manifest their transcriptional and cellular effects until much later, if such regulators function in the memory pathway as opposed to transcription pathway. This is an important consideration for the entire epigenetics field, especially in our current age of acute degradation approaches, and it will encourage better understanding of epigenetic pathways in general.

One issue that needs to be addressed before publication relates to the RNA turnover aspect. In the experiment in Figure 2B this analysis is conducted at 4h not 24h (when the initial memory effect was detailed in Figure 1), so is your conclusion about no role for RNA turnover relevant to the memory phenomenon? Especially in the light of Figure 7, where both the experiment and model indicate a longer timescale is required to get the memory effect.

We agree with the reviewer that this was an important issue to address, and have now included an additional RNA degradation experiment conducted after the 24h recovery process, during the 2^nd^ induction (Figures 2B, Figure 2-supplement 1B). One possibility that we wanted to address in this manuscript is whether lower E74 mRNA levels in Nup98-depleted cells in memory conditions can be explained by higher degradation of the mRNA upon Nup98 KD. As can be seen in the control vs. Nup98 KD conditions, the degradation rate of E74 mRNA during the 2^nd^ induction in either condition is still quite similar to the control 1^st^ induction, allowing us to make the conclusion that changes in RNA turnover do no contribute significantly to the observed memory defects in Nup98-depleted cells. If anything, the E74 mRNA might possess a mildly increased lifetime during the 2nd induction in Nup98 KD relative to WT conditions (mRNA degrades at a slightly lower rate). Although the difference is not large, a potentially lower degradation rate suggests that the transcriptional output in Nup98-depleted conditions during the 2^nd^ induction may be even lower than we report. Furthermore, this minor difference is not reflected in our mRNA export assessments in Figure 2A – the nuclear/cytoplasmic ratio of E74 mRNA remains unchanged by Nup98 KD in either 1^st^ or 2^nd^ inductions (mRNA export can influence mRNA degradation dynamics). Based on these combined results, we conclude that the memory defect (lower levels of E74) upon Nup98 KD cannot be explained by an increase in mRNA turnover.

Reviewer #2:Transcriptional memory is a phenomenon whereby certain inducible genes show stronger/faster induction if they have previously been expressed. This behavior can persist through multiple cell divisions. Such phenomena have been explored in yeast, plants, flies and mammalian cells and several different types of memory have been discovered. In the cases where we have some mechanistic understanding, there are two classes of memory. The regulation of some genes is impacted by the persistence of transcriptional regulators that were previously produced and cytoplasmically inherited (e.g. yeast GAL1). The second class of genes (e.g. yeast INO1 and interferon γ-induced genes in HeLa cells) there are cis-acting changes in transcription factor binding and/or chromatin structure that impact future expression. This latter class of genes shows a physical interaction with the nuclear pore protein Nup98 and memory requires Nup98.This manuscript explores the transcriptional memory of ecdysone-regulated genes in *Drosophila* S2 cells, a phenomenon that is Nup98-dependent and correlates with a Nup98-modulated enhancer-promoter loop. The focus of the study is a thorough and quantitative anlalysis of the population dynamics of transcription during the initial induction and memory phases using single molecule RNA FISH. These data are then used to assess three different quantitative models for memory: 1. A two-state model in which memory increases the number of cells that respond to ecdysone, 2. A two-state model in which memory increases the rate of RNA polymerase loading onto the promoter and 3. A four-state model in which these genes exist in an equilibrium between two different expression states (low and high) and this equilibrium is regulated by memory. The authors find that model 3 is the best fit for the observations and, importantly, this model makes two predictions that they could test. First, this model predicts that transcription should not be required for memory. Second, memory should "develop" following the initial induction over some time scale. The authors then test both of these predictions and find that they are true. The former is interesting and rules out that the production of a transcriptional regulator during the first induction is key to this process. The latter is very interesting and surprising and will stimulate additional work to discover its meaning. Overall, the paper is interesting and impactful and worthy of publication in eLife. I have some suggestions for clarifying changes to the manuscript and, if possible, additional experiments that would strengthen the story.1. The four state model is not well described in the manuscript and the schematic in Figure 5A is confusing. This schematic suggests that there are equilibria between non-memory and memory (which is understandable) and between low-expressing and high-expressing states (also understandable) but that there is also an equilibrium between non-memory and low-expressing and between memory and high-expressing, which does not make sense to me. I think what the authors are trying to convey is that the equilibrium between low-expressing and high-expressing states is influenced by Nup98-dependent memory. If so, then the model should include the following states: non-memory, low-expressing, non-memory high-expressing, memory low-expressing and memory high-expressing. Between each of these states would be an equilibrium and the equilibrium constants for the equilibrium between low and high expressing states would be different for non-memory and memory. I assume that this is how they generated their model. If not, please spend more space to clarify the model and provide an explanation of how the non-memory state is in equilibrium with the low-expressing state, etc.

We apologize for the confusion surrounding our final model and have now substantially reworked the explanations and the terminology, as well as provided a new schematic for it (now in Figure 4A and accompanying text). Generally, our model is based on envisioning 4 states that an ecdysone-responsive gene can be in:

1) a state without ecdysone stimulation and that is not transcribing substantially or is at basal levels (this is the top left state in the diagram, which we now renamed “Uninduced Default” or “UD”) – this state is the default state in non-stimulated conditions (we have added kpolB to describe this very low basal rate of transcription in the absence of ecdysone);

2) a state in conditions of ecdysone stimulation and that is characterized by default transcriptional response, which is relatively low compared to the high states we observe – we renamed this state “Induced Default” or “ID” (this transcriptional response is described by kpolL);

3) a state in conditions without ecdysone stimulation, that is not transcribing above basal levels, but that has established memory – we renamed this state “Uninduced Memory” or “UM” (this is the state that would express highly if exposed to ecdysone);

4) a state in conditions of ecdysone stimulation and that has established memory, which together produce a high transcriptional response – we renamed this state “Induced Memory” or “IM”; (this transcriptional response is described by kpolH).

In this model, a “memory high-expressing” state is now termed IM (used to be termed “Highexpressing”), and there would be no “non-memory high-expressing” state – the high-expressing state is the equivalent of the memory state, in the presence of ecdysone (see Figure 4A). The IM state is the product of a conversion governed by the rate constant kC, combined with ecdysone stimulation (the way to get to this state would be either via kA, then kC process, or via kC, then kA process). Similarly, there would be no “memory low-expressing” state, only a “nonmemory low-expressing” state (“ID”), according to our model.

In basic terms, the right half of the diagram means ecdysone-induced transcription, and the left half means no ecdysone-induced transcription. The un-induced states (or those states with only basal transcription in the absence of hormone) are either non-memory or memory, and this intro-conversion is defined by kC. The induced states (in the presence of hormone) are also either non-memory (low-expressing) or memory (high-expressing), and this intro-conversion is similarly defined by kC. The switch into hormone-induced transcription is governed by kA. We believe that this separation of the kA and kC governed processes is what explains the observed independence of the re-induction response from first induction transcription.

Although we agree that models with additional states and additional rate constants could also work well, we generated our model as the simplest/smallest set of states and parameters that can describe our in vivo data and that can generate testable predictions (that we have tested and provided evidence for).

2. The technical perfection of smRNA FISH is unclear. I doubt it is perfect. Yet, the modeling is highly leveraged on the goal of matching the smRNA FISH data. Some additional controls would strengthen the interpretation. For example, the untreated S2 cells (0h) should be included for the smRNA FISH and modeling. Likewise, knowing how control genes that are not up/down-regulated by ecdysone behave in smRNA FISH would improve the rigor of this approach.

To address the reviewer’s concern, we have now included smRNA FISH with control probes against genes that do not change expression upon ecdysone addition (according to Shlyueva et al. Mol. Cell. 2014) – as shown in Figure 1-supplement 2A-C, the observed distribution of smRNA FISH signal for the mRNAs of such genes Pp4-19C and Rpt6 does not increase or change considerably with addition of ecdysone, unlike that for E74 (Figure 1D-E).

In prior publications, we have shown that our single molecule counting has a maximum error rate of roughly 12% (Little SC et al., 2013, 2015), whereas the error is approximately 4-5% in the quantification of instantaneous transcriptional activity (Zoller B et al., 2018). To further demonstrate the technical accuracy of our smRNA FISH quantification, we now also included further analysis of single diffraction limited spots of our smRNA FISH, as well as representation of average spot intensities for all conditions (Figure 1-supplement 2D-F), providing evidence that the vast majority of smRNA FISH objects are diffraction-limited.

For cells at time 0h, our quantitative RT-qPCR measurements indicate an average production rate of 0.02 mRNAs per minute. Thus, the distribution of CUs at 0h is heavily skewed toward zero and appears to be uninformative in a model of ecdysone-induced activity. For this reason we do not include it in the histograms shown in Figures 3-4, although the production rate at 0h has been included into the fitting of the RT-qPCR data.

3. Some molecular information about transcription factor occupancy or chromatin changes at the enhancer that is involved in the memory-enhancing loop would greatly strengthen the mechanistic insights of this paper.

We very much agree with the reviewer that these questions are very interesting and important in the context of memory-associated enhancer loop. We are currently conducting these types of experiments, as well as testing actual looping in various conditions. We hope to compile this data as a separate manuscript in the future, and believe that given the complexity of the question, this information currently lies outside the scope of this manuscript.

Reviewer #3:In this manuscript by Pascual-Garcia et al., the authors use single-cell gene expression analysis to dissect transcriptional memory in the ecdysone response. They explore several different models and the surrounding parameter space in order to fit the data and then arrive at a plausible multi-state model for how Nup98 might be enabling transcriptional re-activation. The overall conclusion is that ecdysone stimulation in the first round 'primes' the re-activation state through a Nup98-dependent process. This priming does not depend on transcriptional activation because treatment with the elongation inhibitor flavopiridol doesn't interfere with the memory response. Although there is not much biochemical insight, I find this conclusion interesting. Overall, the analysis is fairly rigorous, and I applaud the effort to apply more quantitative models to the field of transcriptional memory.

We are grateful for the reviewer’s positive assessment of our manuscript and for appreciating our effort to describe transcriptional memory with a quantitative model.

1. The modeling is a mix of ordinary differential equation analysis and stochastic simulations with the Gillespie algorithm. It is a little unwieldy at times, and I feel the manuscript spends quite a bit of time on a model which doesn't work. Since this non-working model is the authors' invention (and not a standard in the field per se), it seems a bit awkward to present the story this way. Ideally, all the models would be stochastic models from the beginning since the single-cell analysis is the main thrust at the end. Although I find their approach a little ad hoc given the current state of the field, I don't feel strongly enough about it to demand a more rigorous treatment.

We agree with the reviewer that in the original version, too much of the manuscript was devoted to the models that did not match the in vivo data. We thus restructured our Figures and combined Figures 3-6 into Figures 3 and 4, with the rest of the parameter and modeling description put into Supplementary figures. The revised manuscript offers more focus on the model that did work and we hope this helps address the reviewer’s comment.

2. The persistence of memory in the presence of flavopiridol is a striking result. Would the authors predict a similar result for an inhibitor working upstream of transcriptional pause release? Specifically, the authors mention in the discussion that enhancer-promoter loops may be implicated. One experiment which the authors might consider (which I am not suggesting as necessary for a revision) is whether antagonists or partial agonists can uncouple these two activities of ecdysone. In other words, they have a negative result which emerges as a model prediction but not a positive result.

The question of earlier transcriptional inhibitors is very interesting, and we have addressed it in our revised manuscript. We have now carried out this experiment with an additional inhibitor Triptolide, which is known to inhibit transcription at an earlier, initiation-associated point (Titov et al., Nat. Chem. Bio. 2011), and observed that treatment with Triptolide similarly allows for the majority of the transcriptional memory response to occur (Figure 5B and 5E). Interestingly, we found that Triptolide treatment affected memory formation to a greater extent than Flavopiridol, but still produced around 60-70% of memory remaining (Figure 5E). We have also performed these transcriptional inhibition experiments on additional ecdysone-inducible genes E75 and E23 and detected similar effects in both treatments (Figures 5E and Figure 5-supplement 1C-D). Furthermore, using smRNA FISH, we observed similarly high re-induction levels in Flavopiridol and Triptolide treated conditions by mRNA spot counts (Figure 5C and Figure5supplement 1B). Based on these combined results, we concluded that inhibiting transcription upstream of transcriptional pause release during initial induction similarly allows for transcriptional memory, although at somewhat lower levels than inhibiting transcriptional elongation.

The implication of the enhancer-promoter loop is also interesting, and we thank the reviewer for the suggestion. It would indeed be very informative to find antagonists that can influence the enhancer-promoter loop or influence the two activities of ecdysone separately, and we plan to explore these approaches in our future work.

Reviewer #4:The authors use a combination of RT-qPCR, smFISH and mathematical modeling to probe the mechanism and contribution of NUP98 to transcriptional memory upon hormone stimulation in *Drosophila* cells. The authors reveal that the transcriptional memory response is independent of transcription levels immediately after hormone treatment. Furthermore NUP98's role in transcriptional memory relates to its ability to stabilize a slow transition (~20 hrs) of the gene from low-expression to high-expression state. While the results are quite striking and the interpretations are timely in some regards, a few controls and clarifications in the text are necessary before recommending publication in eLife.1. The authors rely heavily on smFISH to validate their computational modeling. However, the accuracy of quantitation of the smFISH is not evident. The authors should show a histogram of intensities of single diffraction limited spots along with a representative figure of multiple zoomed in panels of the single RNA spots. This is important as the majority of spots, particularly in the transcriptional memory response, are not diffraction limited which complicates both the presentation and analysis of their data.

To address the reviewer’s concern, we now include panels of average single spot intensities for all conditions and further analysis of single diffraction limited spots of our smRNA FISH (Figure 1-supplement 2D-F). Our prior work has shown that the vast majority of smFISH objects in our RNA FISH approach are diffraction-limited, including actively transcribing loci, regardless of mRNA content, and that such objects can contain the equivalent of hundreds of mature mRNAs (Little SC et al. 2013, Petkova MD et al. 2014, Little SC et al. 2015, Zoller B et al. 2018). The same is true for the objects detected in this study, where the full width at half-maximum (FWHM) of the detected objects is identical regardless of their intensities, as we now show in Figure 1supplement 2D-F. The FWHM (110 nm) is more or less the same as the radius expected from the Abbe limit (emission wavelength / numeric aperture = 105 nm) regardless of how bright the spots become. So the spots are all the same apparent size and their size is that expected for diffraction-limited objects.

2. Is there any evidence of E74 being alternatively spliced which may differentially affect FISH probe binding to the mRNAs producing large heterogeneously sized diffuse spots seen in the figure 1D?

The E74 locus does produce two transcript isoforms – the long isoform, transcribed from its main promoter, and the short isoform, transcribed from an alternative promoter within the gene. The two mRNA isoforms share 5 exons (Author response image 1). Our previous work (Pascual-Garcia et al. Mol Cell 2017) and current RT-qPCR analysis found that it is the long isoform that changes in response to ecdysone and exhibits transcriptional memory in S2 cells (Author response image 1), thus we designed our smRNA FISH probe to favor the long probe: the smRNA FISH probes were designed to the 5 exons, excluding the short isoform-specific exon (Author response image 1). With this design, if RNA pol II (green) distributes uniformly along the length of the gene (Author response image 1), the number of probes bound to a nascent mRNAs attached to a transcribing RNA pol II increases with total transcribed distance. Therefore probes binding 5' sequences are represented more frequently than those binding 3' sequences, as illustrated by the cyan probes in the diagram (6 sites for the 5' cyan probe but only 3 for the 3' cyan probe, Author response image 1). Thus, there is a strong bias towards the 5’ long form-specific exons in detection by these probes, as shown in Author response image 1 – provided uniform distribution, we can estimate that the long isoform would account for 95% of the probe signal.

**Author response image 1. sa2fig1:** Utilized smRNA FISH probe detects mainly the long isoform of E74.

RNA pol II does not have to distribute uniformly along the gene, but our single-cell FISH data supports the predominance of the long isoform in our detection – if we compare dynamics of the FISH-detected transcription sites with the RT-qPCR-detected dynamics of the long isoform vs. a mixture of the long and short isoforms, we observe a much better correlation to the long isoform only (Author response image 1).

<inline-graphic mime-subtype="png" mimetype="image" xlink:href="media/image1.png" />Furthermore, even if we assume that the rate of transcription from the alternative/short promoter is 10X that of the main/long promoter, given the 5’ bias in detection described above (Author response image 1), we can estimate that the fraction of the short isoform in the total mRNA is still negligible for our analysis (Author response image 1). In such an estimation based on our isoform-specific RT-qPCR, we see that the only times when the amount of short isoform approaches that of the long isoform are at Uninduced and Recovery time points (Author response image 1). At these time points, we did not observe easily detectible nascent transcription sites by smRNA FISH, and these time points were not used in our modeling for this reason (more precisely, the uncertainty of assigning nuclear FISH dots to being a transcription site is too high at these uninduced time points.) At the time points that were used in our modeling, the fraction of the short isoform is so small (for example, Author response image 1, 2nd induction – less than 1%) that we can ignore it. The approximately 10% additional signal during the 1st induction is less than the typical error of signal intensity (Little et al., Cell 2013). Thus at worst, we are overestimating the output during the first induction by about 10%, but this still falls well within the confidence intervals we report (Figures 2-4) and thus does not change the conclusions or the numbers that we present in the paper.

We did not include this analysis into the manuscript, but can do so if the reviewer feels that we should.

3. While the differences are readily apparent in most of the data, the authors do not show any statistical testing in many if not most plots (e.g. violin plots Figure 1E). This should be corrected in a revised manuscript.

Per reviewer’s suggestions, we have now added statistical testing and resulting p-values for Figure 1E (now in Figure 1-supplement 1D, F) and have also included statistical assessments into Figure 2. For model fittings in Figures 2C-E, 3 and 4, we also provide 95% confidence intervals that describe the probable range of the fitted rates (Figure supplements for Figures 3, 4).

4. S2 cells are considered aneuploid. For nascent transcription site analysis, it would be helpful for the readers to see a simple diagram showing the percentage of cells showing 1, 2, 3, 4 etc active TS sites throughout the time course as it was not apparent whether a certain percentage of cells in the population never expressed E74.

In response to reviewer’s concern, we would like to mention several points. First, S2 cells, like most *Drosophila* cell types, have a strong tendency to align/cluster their homologous chromosomes in interphase. As a result, by direct observation we most often observe a tight cluster of diffraction-limited spots by DNA or RNA FISH, although we can also observe a fraction of cells with multiple physically separate sites by either DNA or RNA FISH (FISH studies by Eric Joyce’s lab have suggested that chromosome 3L, which contains the E74 locus, is hexaploid in S2 cells).

Second, the way we designate active transcription sites and measure transcription in our imaging analysis makes it difficult to directly assess the fraction of cells that do not transcribe E74, but allows us to estimate it – this estimate is always very small, as detailed below:

We designate active sites using a threshold of three times the intensity of single mRNAs. This allows us to estimate the fraction of loci that have not transcribed sufficient immature RNA to allow detection. This fraction is naturally an underestimate of the true fraction of loci that have not been activated. To address this underestimate, we developed a method in prior studies (Little et al., Cell 2013, Zoller et al., Cell 2018), where we have examined the consequences of this underestimate by first identifying cells with fewer than the expected number of detected loci, then assigning to those cells additional candidate loci by selecting intra-nuclear spots either at random or by choosing the brightest available intra-nuclear object. In our current studies, we have verified that the objects chosen by either method are relatively dim. The relative uniformity of the dim intensities means that neither method outperforms the other when used in the modeling. We therefore conclude that our estimates of the activation rate kA = 0.017 +/- 0.001 is accurate. We can therefore calculate the fraction of inactive loci as a function of time as I = exp(-t*kA). At 1, 2, and 4 hrs we find that 36%, 13%, and 1.7% of loci remain inactive. Because of the presence of multiple loci per cell, we find that the fraction of cells with zero active loci is always very small: 0.2%, 5x10^-6^, and 2x10^-11^.

5. Figure 7 is one of the most key figures that the authors use to conclude that NUP98 is stabilizing a memory state independent of transcription. Yet they have chosen to use a less informative assay (e.g. RT-qPCR) rather than the most sensitive assay (e.g smFISH) to support their model and conclusions. The authors would make a stronger case for their findings if they employed smFISH to monitor transcript abundance and TS activity after inhibiting transcription.

To address the reviewer’s point, we have now carried out smRNA FISH assay against E74 in conditions of transcriptional inhibition by either Flavopiridol or Triptolide (now in Figure 5C and Figure 5-supplement 1B), and similarly observed high levels of transcriptional reinduction/transcriptional memory by mRNA spot counts. We thank the reviewer for suggesting this experiment, as it does provide for a stronger case for our conclusions.

6. FP primarily only prevents Pol II pause escape and not promoter escape of Pol II. So transcription initiation and Pol II loading at the promoter still occurs in flavopiridol treated cells. This is important because chromatin marks associated with active transcription are likely still being placed in the promoter and enhancer regions of E74 even though a complete transcript is not being produced. The authors should repeat their assays and treat the cells with inhibitors such as triptolide or THZ1 that act earlier in the transcription cycle or better yet use an ecdysone receptor mutant that prevents Pol II recruitment or chromatin modification to the gene in the first place.

We thank the reviewer for the excellent suggestion, and have now included an additional experiment, which used Triprolide to inhibit transcription during the initial treatment with ecdysone (new Figure 5D-E). As the reviewer points out, Triptolide acts earlier in the transcription cycle, as an inhibitor of TFIIH (Titov et al., Nat. Chem. Bio. 2011). We observed that treatment with Triptolide similarly allows for the majority of the transcriptional memory response to occur (Figure 5B and 5E). Interestingly, we found that Triptolide treatment affected memory formation to a greater extent than Flavopiridol, but still produced around 60-70% of memory remaining (especially as compared to much stronger effects of Nup98 KD and shorter recovery – Figure 5E). We have also performed these transcriptional inhibition experiments on additional ecdysone-inducible genes E75 and E23 and detected similar effects in both treatments (Figures 5E and Figure 5-supplement 1C-D). Furthermore, using smRNA FISH, we similarly observed high re-induction levels in Flavopiridol and Triptolide treated conditions by mRNA spot counts (Figure 5C and Figure 5-supplement 1B).

Based on these combined results, we concluded that inhibiting transcriptional initiation during initial induction similarly allows for transcriptional memory, although at somewhat lower levels than inhibiting transcriptional elongation. We have added a discussion of these points into the main text.

[Editors' note: further revisions were suggested prior to acceptance, as described below.]

All reviewers find that the manuscript is novel and exciting. However reviewer #4 has additional comments that need to be addressed, in particular the observation 1 that triptolide is not required for PIC formation in *Drosophila*. This issue needs to be addressed in the Discussion along with other issues raised by this reviewer. Please send a response to these comments with the revised manuscript.

We sincerely thank the reviewers and the reviewing and senior editors for the input on the manuscript and for the positive assessment of our work. As outlined below, we have now addressed the remaining comments from reviewer #4 and included all the needed changes in the re-revised manuscript.

Reviewer #4:The authors take an important step in quantitatively assessing transcriptional memory in the context of mathematical modeling and testing of the models through single cell approaches. Furthermore, the authors extend their work to show how single cell data relates to population-level transcription outcomes. This second approach is highly relevant since transcriptional memory has been traditionally studied at the population-level allowing interpretation of previous studies in the context of single cell data. More importantly, the models produced can make predictions that the authors successfully test to show that transcription initiation is not necessary for establishment of memory. The authors do a good job of addressing the reviewer's critiques. While a clear mechanism for Nup98 establishment of memory is still lacking, I think that this manuscript represents an important step in integrating single cell data into studying memory. There are a few critiques that should still be addressed before final acceptance for publication.

We thank the reviewer for the overall positive assessment of our revisions and suggestions to improve the manuscript. As detailed below, we have incorporated all of the latest suggestions and resolved the remaining issues raised by the reviewer.

1. While the authors indeed show that transcription initiation is not required for memory, Triptolide treatment does not prevent PIC formation in *Drosophila* (Krebs et al., 2017, Mol Cell, which the authors cite in the manuscript). PIC assembly actually increases during Triptolide treatment suggesting that key factors and histone marks related to the assembly of the PIC (e.g. H3K4 methylation due to its presumed role in recruiting TFIID) are likely still present and capable of establishing memory in the absence of initiation. The authors should be very clear of this possibility in the discussion.

We thank the reviewer for pointing out this possibility and agree that this is an important point to add. As the reviewer states, while our experiments demonstrate that transcriptional memory can still occur in Triptolide-inhibited conditions, they do not delve into which of the ecdysone-triggered initiation events that still occur in the presence of Triptolide are required for this preservation. PIC assembly or setting up permissive chromatin structure are indeed likely candidates for establishing memory. We have now included this point into the revised discussion (first paragraph of Discussion section).

2. Lines 175-177 the authors are essentially making a statement about transcriptional noise being independent of Nup98. It would be highly beneficial to quantitatively measure transcriptional noise (CV2 or std2/mean2) under the different conditions to validate this statement.

Per reviewer’s suggestion, we now provide CV2 measurements of transcriptional noise under different conditions in Figure 1-supplement 2G, plotted as noise (CV2) as a function of mean number of mRNA molecules per cell. These plots show similar trends between the different conditions, suggesting that Nup98 does not contribute substantially to transcriptional noise.

3. I realize that Models 1 and 2 are not viable in the end. However, the wording in Lines 301-306 is misleading creating a potential flaw in these models that seems to be carried throughout the manuscript. The authors invocation of a quantitative assessment of the PIC (e.g. Author's should use the term PIC since RNA Pol II holoenzyme can be confusing) footprint into the model is naive and underestimates the role of a stable TFIID scaffold and TFIIB in re-initiation rates (Zhang et al. Genes and Dev 2016, Yudkovsky et al., Nature 2000). It is highly likely that core promoter accessibility plays a major role in going from an inactive to active state. But core promoter accessibility would likely be a binary effect and not scaleable as the authors seem to intimate in lines 302-304 and let freely fluctuate in their modeling. Model 1 and 2 also assumes that transitions to the inactive states are negligible suggesting that the gene is stuck in the ON state and therefore Nup98 is enhancing Pol II's attempted loading onto a stable PIC scaffold of defined size (e.g. footprint).

We thank the reviewer for altering us to the overly simplistic and potentially misleading presentation of Model 2. As previously written, the model (1) ignores "bursts" of activity resulting from promoter state (e.g. accessibility) switching, and (2) appears to use a "naive" estimate of the Pol II footprint on DNA. We have now rewritten the presentation of Model 2 (second to last paragraph of section 3 of Results) to emphasize that the quantity that matters is not the footprint per se, but is instead the interval between sequential Pol IIs transcribing a given nucleotide.

It was most straightforward to implement this model by allowing the Pol II footprint to vary as a free parameter, as we noted in the prior version of the main text, combined with a constant elongation rate of 1500 nt/min, a value supported by multiple studies in *Drosophila*. Use of this parameter, paired with the "loading acceleration" parameter, permitted us to fit the qPCR data. The model is thus agnostic about the nature of the process(es) that determine the minimum interval. We note that the maximum average loading rate kPol is ~3 mRNA per locus per minute in model 2. This is less than half of the maximum that has been observed in *Drosophila* (e.g. Zoller et al. 2018).

We believe it is likely that promoter switching in this context prohibits achievement of the maximum rate observed in embryos. We also note that the promoter switching rates would need to be rapid in order to support this rate of mRNA production. While such switching could be assessed using multi-color FISH (as in Zoller et al. 2018), we have not as yet produced such data.

4. It would be interesting to see whether the FP and TL treatments change the distributions of transcriptional output during the second induction in single cells compared to those shown in Figure 4.

To address the reviewer’s point, we have compared the population distributions of transcriptional output in FP or TPL treated conditions of 2^nd^ induction, relative to control 2^nd^ induction, using the smRNA FISH data in Figures 5 and 4 (the comparison is now in Figure 5 supplement 1C). The distributions do exhibit some minor differences, the extent of which and the mechanism behind which could be subjects of future studies.